# Engineering a synthetic pathway for maleate in *Escherichia coli*

Shuhei Noda[1], Tomokazu Shirai[1], Yutaro Mori[1], Sachiko Oyama[1] & Akihiko Kondo[1,2,3]

Maleate is one of the most important dicarboxylic acids and is used to produce various polymer compounds and pharmaceuticals. Herein, microbial production of maleate is successfully achieved, to our knowledge for the first time, using genetically modified *Escherichia coli*. A synthetic pathway of maleate is constructed in *E. coli* by combining the polyketide biosynthesis pathway and benzene ring cleavage pathway. The metabolic engineering approach used to fine-tune the synthetic pathway drastically improves maleate production and demonstrates that one of the rate limiting steps exists in the conversion of chorismate to gentisate. In a batch culture of the optimised transformant, grown in a 1-L jar fermentor, the amount of produced maleate reaches 7.1 g L$^{-1}$, and the yield is 0.221 mol mol$^{-1}$. Our results suggest that the construction of synthetic pathways by combining a secondary metabolite pathway and the benzene ring cleavage pathway is a powerful tool for producing various valuable chemicals.

---

[1] Center for Sustainable Resource Science, RIKEN, 1-7-22, Suehiro-cho, Tsurumi-ku, Yokohama, Kanagawa 230-0045, Japan. [2] Department of Chemical Science and Engineering, Graduate School of Engineering, Kobe University, 1-1 Rokkodai, Nada, Kobe 657-8501, Japan. [3] Graduate School of Science, Technology and Innovation, Kobe University, 1-1 Rokkodai, Nada, Kobe 657-8501, Japan. Correspondence and requests for materials should be addressed to A.K. (email: akihiko.kondo@riken.jp)

Bioprocesses using conventional microorganisms such as *Escherichia coli* and yeasts have been widely studied in the past few decades[1–3]. These clean processes using renewable feedstocks as the raw materials are expected to provide solutions to the problems of global warming and exhaustion of fossil fuels. Metabolic engineering enables the construction of synthetic pathways for the production of valuable compounds using metabolically optimised microorganisms. Using rational approaches based on metabolic engineering and synthetic biology, various genetically modified microbes have been created to produce fuels, bulk chemicals, amino acids and pharmaceuticals[4–6].

Maleic acid is one of the most valuable organic acids, and its derivative, maleic anhydride, can be converted into various polymer materials and pharmaceuticals[7–10]. Maleic anhydride is easily obtained by heating maleic acid at 135 °C. The market volume for maleic anhydride was estimated to have reached 1,807,000 tons in 2007, which is much larger than the estimated volumes of other four-carbon dicarboxylic acids, such as succinic acid and fumaric acid (270,000 and 90,000 tons per year, respectively)[11]. Maleic anhydride is chemically synthesised via catalytic oxidation of *n*-butane in the presence of vanadium phosphorus oxide under high temperature (>400 °C)[12]. Despite the high-global demand for maleic acid and maleic anhydride, there have been no reports on microbial production of maleic acid using renewable feedstocks.

Organic acids, including succinic acid, fumaric acid, malic acid and 3-hydroxypropionic acid, have been categorised as a key group among the top platform chemicals[13–17]. Succinic acid can be easily converted to other valuable chemicals, such as 1,4-butanediol, and used for the production of the biodegradable plastic polybutylene succinate[18]. Fumaric acid is a starting unit for polymerisation and esterification reactions in the production

of unsaturated polyester resins and food or beverage additives[7, 15]. Production of these organic acids is usually achieved in microorganisms via the tricarboxylic acid (TCA) cycle. Some other organic acids, including 3-hydroxypropionic acid, malic acid and adipic acid, are also mainly synthesised via the TCA cycle[14, 19–21]. Thus, extending the TCA cycle is the most common strategy to produce industrially important organic acids using microorganisms.

Maleic acid is formed in the benzene ring cleavage pathway, which contributes to the degradation of environmental pollutants by some microbes[22, 23] (Fig. 1). In this pathway, aromatic chemicals such as naphthalene, cresol and xylenol are converted via several steps to 3-hydroxybenzoate (3HBA) and gentisate (2,5-dihydroxybenzoic acid), which are the key intermediates of the pathway. Then, 3HBA is converted to gentisate by 3-hydroxybenzoate 6-hydroxylase (3HB6H). Subsequently, maleylpyruvate synthase (MPS) catalyses the oxidative cleavage of the benzene ring of gentisate to form maleylpyruvate, followed by the degradation to maleic acid and pyruvate or isomerisation to fumarylpyruvate and subsequent degradation to fumaric acid and pyruvate[22, 23] (Fig. 1). However, microorganisms capable of degrading aromatic compounds do not have the ability to endogenously produce gentisate from sugars or other renewable carbon sources via the central metabolic microbial pathways such as the glycolysis pathway or pentose phosphate pathway.

Some terpenoids and polyketides produced by several *Streptomyces* species, as well as by other bacteria, are known to include the 3HBA moiety, which is a significant starter unit in the biosynthetic pathway[24–26]. 3HBA synthase generates 3HBA from chorismate, which is an important intermediate in the synthesis of aromatic amino acids in microorganisms. Although several compounds, such as aromatic amino acid and benzoate

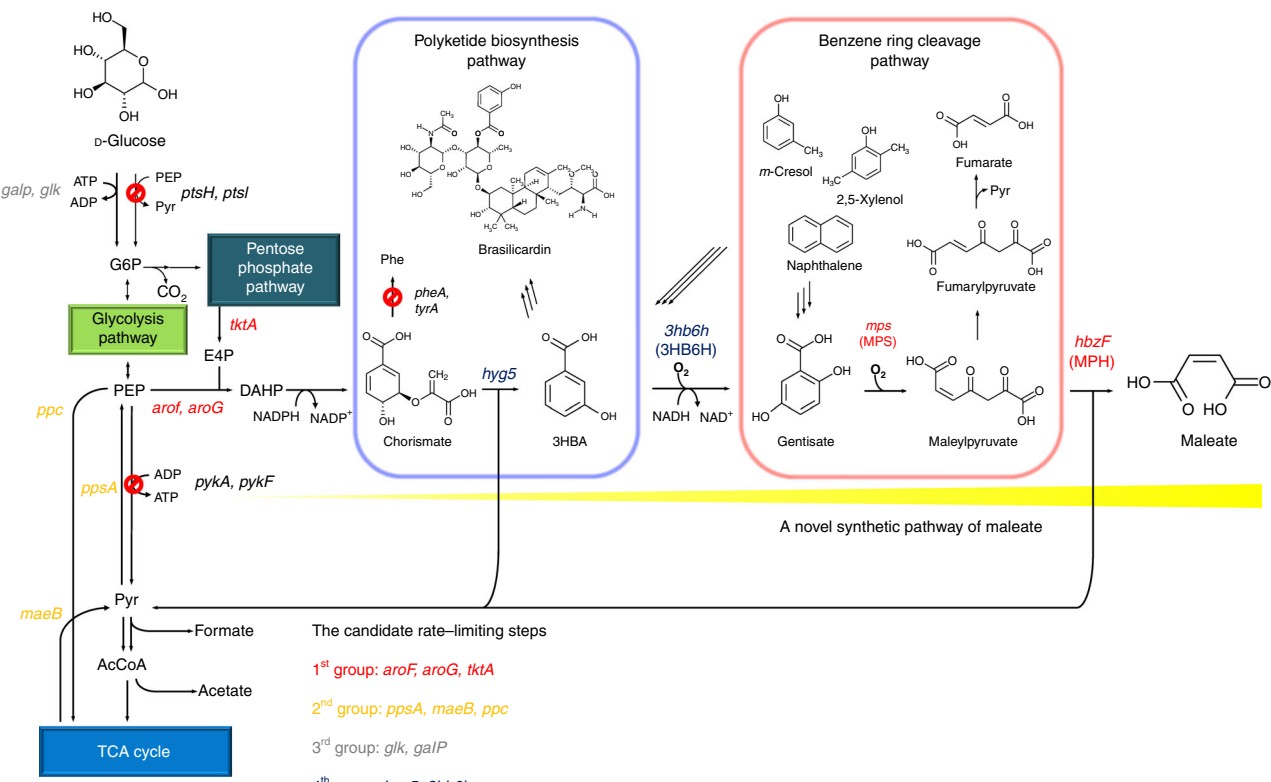

**Fig. 1** The synthetic metabolic pathway for the production of maleate in *E. coli*. AcCoA acetyl coenzyme A, Cho chorismate, DAHP 3-deoxy-D-heptulosonate-7-phosphate, E4P erythrose-4-phosphate, G6P glucose-6-phosphate, 3HBA 3-hydroxybenzoate, (Iso)Cho isochorismate, Mal malate, NADP+ oxidised NADP, Oxa oxaloacetate, PEP phosphoenolpyruvate, Phe L-phenylalanine, Pyr pyruvate

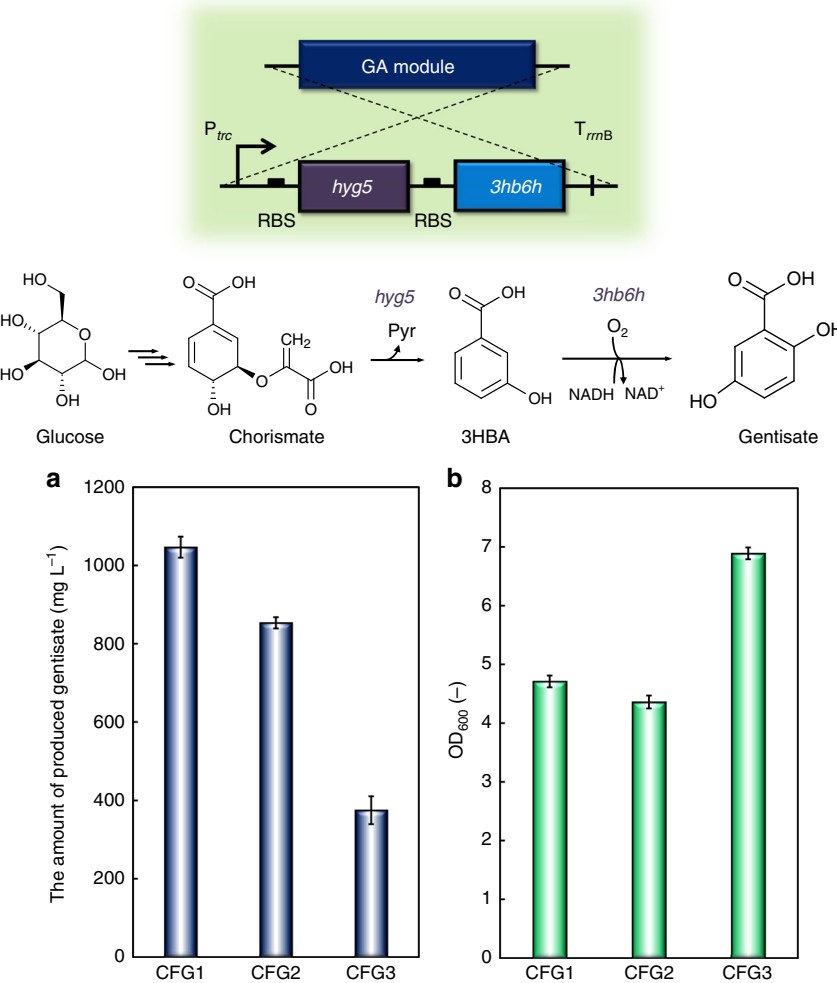

**Fig. 2** Culture profiles of CFG1–3 after 72 h of cultivation. **a** The amount of produced gentisate. **b** $OD_{600}$ values. Data are presented as the mean ± standard deviation of three independent experiments ($n = 3$)

derivatives, have been produced via the chorismate pathway, there have been only few reports on the production of 3HBA or its derivatives using genetically engineered microbes[27].

In the present study, we attain the production of industrially valuable maleate using genetically engineered *E. coli*. We aim to produce maleate by extending the chorismate pathway, whereas many other important dicarboxylic acids, such as fumarate and succinate, are produced via the TCA cycle. First, we focus on the synthetic pathways of gentisate from chorismate and maleate from gentisate, which originate from the polyketide biosynthesis pathway in *Streptomyces* species and the benzene ring cleavage pathway in aromatic compound-degrading bacteria, respectively. Then, we construct a synthetic pathway to produce maleate from simple carbon sources using a previously reported chorismate-overproducing *E. coli* strain[27]. By optimising batch culture conditions in a 1-L jar fermentor through the alteration of the amount of dissolved oxygen (DO), we successfully produce maleate with high productivity, compared to the initially tested condition. Although a maleate synthetic pathway is successfully constructed in *E. coli* and the maleate production drastically increased, the rate and level of production are lower than those reported for other compounds produced by genetically engineered microbes. In the synthetic pathway of maleate created here, 2 mol of pyruvate are also produced. Recycling of pyruvate to maleate would be one of necessary steps to further increase the maleate production for industrial application in the future.

## Results

**Creation of a synthetic pathway for gentisate in *E. coli*.** To construct a synthetic pathway for maleate production from renewable carbon sources, the first step is the reaction converting chorismate to 3HBA (Fig. 1). In our previous report[27], we successfully produced 3HBA from glucose using genetically engineered *E. coli*, and the concentration of 3HBA reached more than $2\,g\,L^{-1}$ in the test tube culture. To create an *E. coli* strain capable of producing gentisate from glucose, we used the 3HBA-producing *E. coli* strain which is the derivative strain of CFT5. Our base strain, CFT5, is a modified *E. coli* strain, with the genes involved in the reaction of conversion of phosphoenolpyruvate (PEP) to pyruvate completely inactivated. In this strain, the phosphotransferase (PTS) system was replaced with a combined system of galactose permease (GalP) and glucokinase (Glk), while the two genes encoding pyruvate kinase (*pykF* and *pykA*) were completely inactivated[27].

Gentisate is synthesised from 3HBA via a reaction catalysed by 3HB6H (Fig. 1). Here, we focused on the following three candidate genes that encode 3HB6H: *cgl3026* from *Corynebacterium glutamicum*, *3hb6h* from *Rhodococcus jostii* RHA1, and *xlnD* from *Pseudomonas alcaligenes* NCIMB 9867. Each gene encoding 3HB6H was introduced downstream of the *hyg5* gene, which encodes the enzyme converting chorismate to 3HBA (Supplementary Fig. 1a). Figure 2 shows the amount of produced gentisate and the cell growth of each gentisate-producing strain

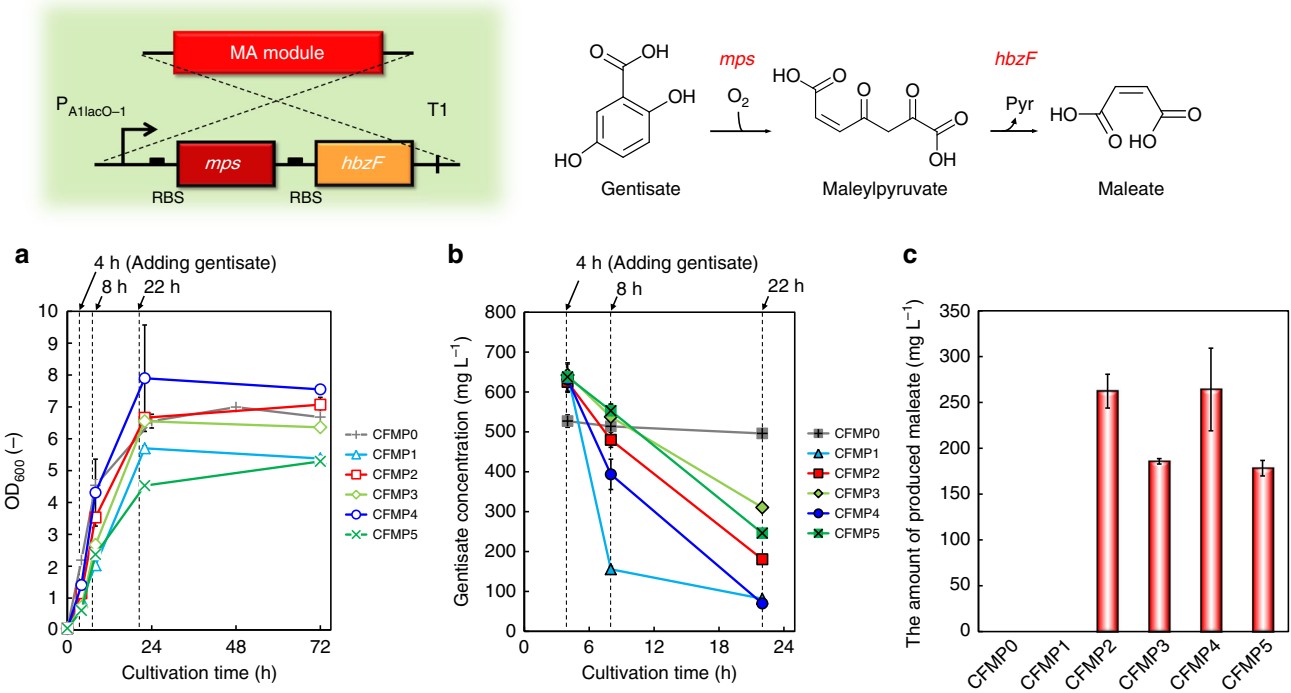

**Fig. 3** Culture profiles of CFMP0–5 in M9Y medium supplemented with 500–600 mg L$^{-1}$ gentisate. Time courses of **a** bacterial cell growth and **b** gentisate consumption. **c** The amount of produced maleate after 22 h of cultivation. Data are presented as the mean ± standard deviation of three independent experiments ($n = 3$)

after 72 h of cultivation. The amount of gentisate produced by CFG1, carrying *cgl3026*, was 1050 mg L$^{-1}$, which was the highest level of gentisate production among the three transformants. CFG2 carrying *3hb6h* and CFG3 carrying *xlnD* produced 850 and 375 mg L$^{-1}$ gentisate, respectively. Although CFG3 showed the highest cell growth, the gentisate production was lower than that in the other two strains. The highest yield of gentisate was 0.165 mol mol$^{-1}$ in the culture of CFG1, carrying the *hyg5* and *cgl3026* gene set, while the yield in the culture of CFG2, carrying *hyg5* and *3hb6h* gene set, was 0.130 mol mol$^{-1}$ (Supplementary Table 1). Based on these results, *cgl3026* and *3hb6h* were selected for subsequent experiments. The cassette containing *hyg5* and a gene encoding 3HB6H under control of the *trc* promoter, used to synthesise gentisate, was referred to as a GA module (Supplementary Fig. 1a).

**Selection of MPS to produce maleate from gentisate**. After the synthetic pathway for the production of gentisate, which is one of the intermediates in maleate production, was successfully constructed, the next step was to confer the gentisate-degrading ability to *E. coli*. Gentisate is degraded to maleate via sequential reactions catalysed by MPS and maleylpyruvate hydrolase (MPH) (Fig. 1).

In the present study, we screened several genes encoding MPS, while *hbzF* from *P. alcaligenes* NCIMB 9867 was adopted as the gene encoding MPH[28, 29]. Using M9Y medium supplemented with gentisate (500–600 mg L$^{-1}$), the gentisate-degrading ability of each strain (CFMP1–5) was evaluated. In this experiment, the additional gentisate was added to the medium after 4 h cultivation. Figure 3 shows the cell growth, gentisate concentration and amount of produced maleate. Gentisate degradation was confirmed for all candidate MPSs tested in this study, whereas CFMP0 carrying the empty vector did not show gentisate degradation (Fig. 3b). CFMP2 carrying *mps* from *Pseudomonas putida* (*sgp1*) and CFMP4 carrying *mps* from *Rhodococcus* sp.

strain NCIMB 12038 (*mps2*) produced 262 and 264 mg L$^{-1}$ maleate, respectively. Surprisingly, CFMP1 carrying *mps* from *Pseudaminobacter salicylatoxidans* (*mps0*) assimilated gentisate more quickly than the other strains; however, the maleate production was not confirmed (Fig. 3c), which indicates that in this strain, gentisate may not be degraded via maleate. In further experiments, we used all tested *mps* genes, except *gdps*, to construct a synthetic pathway of maleate production. Thus, genes involved in maleate production from gentisate were successfully selected. The cassette containing a *mps* gene and *hbzF* under control of the $P_{A1lacO-1}$ promoter, used to convert gentisate to maleate, was referred to as an MA module (Supplementary Fig. 1b).

**Tolerance of *E. coli* to maleate**. Maleate is not usually produced in microbial metabolic pathways, unlike fumarate, which is the geometric isomer of maleate and one of the important compounds in the TCA cycle. Therefore, the tolerance of *E. coli* to maleate was tested. Using a medium with different concentrations of maleate (0–6 g L$^{-1}$), the cell growth of CFT5 was evaluated. Figure 4a, b shows the cell growth of CFT5 and the time courses of consumed glucose, respectively, in the medium with added maleate. As the initial concentration of maleate increased, the maximum optical density (OD$_{600}$) and the initial growth rate slightly decreased. The specific growth rate (μ) in the cultures with 0, 0.5, 1.5, 3 and 6 g L$^{-1}$ maleate was 0.69, 0.64, 0.64, 0.64 and 0.65 h$^{-1}$, respectively, between 2 and 4 h of cultivation. We also investigated whether *E. coli* assimilates maleate. Figure 4c shows the maleate concentrations at 0, 24 and 48 h of cultivation. Although the amount of maleate that was added to the medium only slightly decreased at any initial concentration, maleate did not seem to be a lethal agent for the *E. coli* strain. In addition, no significant change was observed in the maleate concentration, indicating that the *E. coli* strain does not have the ability to assimilate maleate. These results suggested that *E. coli* was a

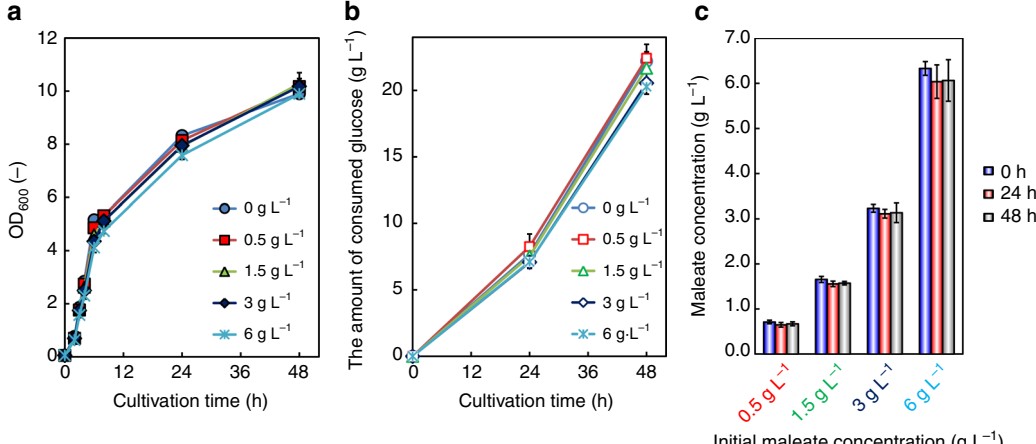

**Fig. 4** Culture profiles of CFT5 cultured in M9Y medium supplemented with different concentrations of maleate. Time courses of **a** bacterial cell growth and **b** the total amount of consumed glucose. The amounts of supplemented maleate were 0, 0.5, 1.5, 3 and 6 g L$^{-1}$. **c** Changes in the maleate concentration during cultivation. Data are presented as the mean ± standard deviation of three independent experiments ($n = 3$)

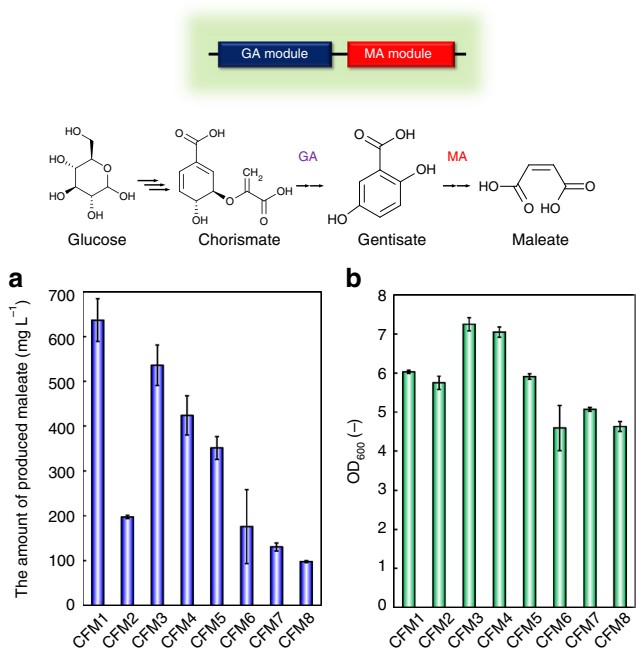

**Fig. 5** Growth and maleate production by strains CFM1–8 after 48 h of cultivation. **a** The amounts of produced maleate and **b** OD$_{600}$ values. Data are presented as the mean ± standard deviation of three independent experiments ($n = 3$)

suitable candidate host for maleate production from renewable feedstocks.

**Creation of a synthetic pathway for maleate from glucose**. As shown in the above sections, gentisate-producing *E. coli* strains and those degrading gentisate to maleate were successfully created. We then continued with the construction of a synthetic pathway of maleate from glucose. First, four different MA modules (including *sgp1*, *mps1*, *mps2* and *mps3*, respectively) were combined with the two different GA modules (containing *3hb6h* and *cgl3026*, respectively), yielding eight different transformants. The MA and GA modules were tandemly cloned into pTrcHisB (Supplementary Fig. 1c). The resultant strains, CFM1–8, were

independently cultured in M9Y medium, and Fig. 5 shows the amounts of produced maleate and the cell growth after 48 h of cultivation. The maximum level of produced maleate, reaching 640 mg L$^{-1}$ after 48 h of cultivation, was achieved using CFM1, which expresses *hyg5*, *3hb6h*, *gdrs* and *hbzF*.

To confirm the maleate production, a culture supernatant of CFM1 was analysed by gas chromatography–mass spectrometry (GC–MS). Specific peaks derived from maleate–bis(trimethylsilyl) ester derivatives ($m/z = 245$ and 147, respectively) were observed at ~3.2 min in the GC–MS spectra of the culture supernatant of CFM1; however, those were not detected in the culture supernatant of the control strain, CFM0, carrying the empty vector (Supplementary Fig. 2). Thus, we successfully constructed a synthetic pathway to produce maleate from glucose via the chorismate pathway.

The amounts of consumed glucose and the yields of maleate obtained after 48 h of cultivation of the eight transformants are summarised in Supplementary Table 2. The highest yield of maleate produced from carbon sources was 0.189 mol mol$^{-1}$ in the culture of CFM1 after 48 h of cultivation. These results indicated that the combination of *hyg5*, *3hb6h*, *gdrs* and *hbzF* would be suitable for maleate production using our base strain CFT5, whose carbon flux to chorismate is increased. Culture profiles of CFM1 were further investigated and are shown in Fig. 6a, b. The maximum level of produced maleate (880 mg L$^{-1}$) was detected after 144 h of cultivation. To evaluate whether other by-products were formed, we assessed the amounts of produced organic acids (Supplementary Table 3) and found that 540 mg L$^{-1}$ acetate and 20 mg L$^{-1}$ formate were produced in addition to maleate. Other organic acids such as lactate and succinate were not detected in this experiment.

**Modular optimisation of the synthetic pathway of maleate**. To improve the production of maleate, we tried to optimise the expression of genes involved in maleate production. First, we replaced the $P_{A1lacO-1}$ promoter in the MA module on pT2c101 with the *trc* promoter. Although the maleate production was observed using a transformant carrying this vector, its level did not increase (Supplementary Fig. 3). Next, the gene sets controlling the production of gentisate from glucose and degradation of gentisate to maleate were tandemly cloned downstream of the *trc* promoter to generate GAMA and MAGA modules (Supplementary Fig. 1d). Figure 6a, b shows the culture profile of CFMt1 carrying the MAGA module. The maximum level of produced

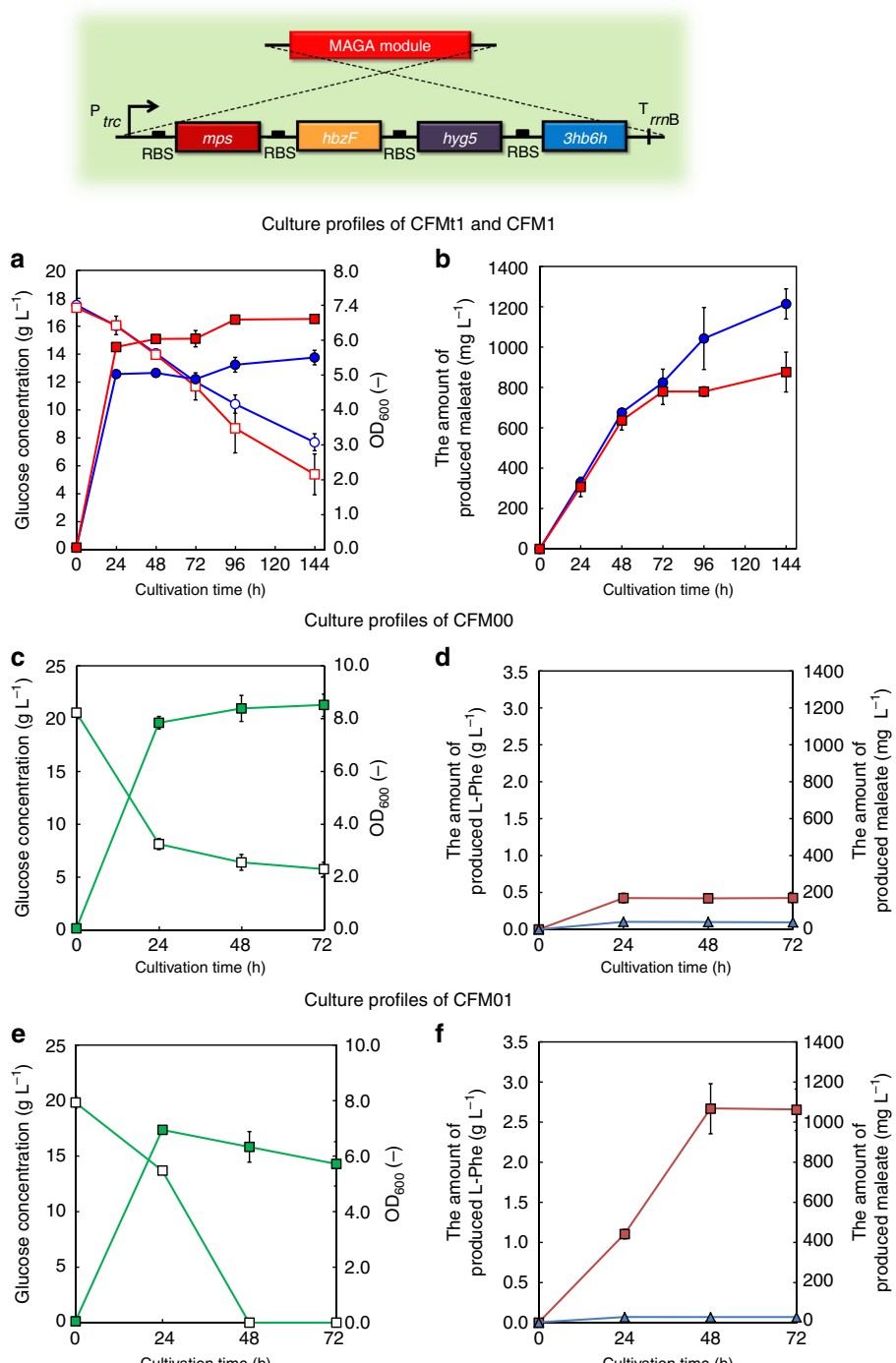

**Fig. 6** Culture profiles of various maleate-producing transformants. Time courses of **a** the bacterial cell growth and glucose concentration, and **b** the amount of produced maleate in the cultures of CFM1 (squares) and CFMt1 (circles). Time courses of **c** the bacterial cell growth (filled symbols) and glucose concentration (open symbols), and **d** the amount of produced maleate (triangles) and L-phenylalanine (squares) in the culture of CFM00. Time courses of **e** the bacterial cell growth (filled symbols) and glucose concentration (open symbols), and **f** the amount of produced maleate (triangles) and L-phenylalanine (squares) in the culture of CFM01. Data are presented as the mean ± standard deviation of three independent experiments ($n = 3$)

maleate (1210 mg L$^{-1}$) was detected after 144 h of cultivation, whereas CFMt2 carrying the GAMA module did not produce maleate (Supplementary Fig. 4). The amounts of produced 3HBA and gentisate were also evaluated, but these compounds were not detected in the culture supernatant of CFMt1. The β-D-1-thio-galactopyranoside (IPTG) concentration used was optimised at 0.1 mM in our preliminary experiments (Supplementary Fig. 5). In the culture of the CFMt1 strain, 640 mg L$^{-1}$ acetate and 18 mg

L$^{-1}$ formate were also produced (Supplementary Table 3). The yield of maleate produced from carbon sources during the entire cultivation was 0.164 mol mol$^{-1}$ (Table 1).

In the present study, we used ATCC 31882 and CFT3 as control strains to demonstrate the usefulness of CFT5. ATCC 31882 is an L-phenylalanine-overproducing *E. coli* strain, and CFT3 is the same as CFT5, except that the former expresses the *pheA* and *tyrA* genes involved in L-phenylalanine production. These two strains

**Table 1 Summary of maleate production by each engineered *E. coli* strain**

| Strain | $P_{max}$ (mg L$^{-1}$) | Maleate yield (mol mol$^{-1}$) | Glucose uptake rate$^{(0-48 h)}$ (mg L$^{-1}$ h$^{-1}$) | Maleate production rate$^{(0-48 h)}$ (mg L$^{-1}$ h$^{-1}$) |
|---|---|---|---|---|
| CFM00 | $40.8 \pm 4.3^{(24 h)}$ | $0.0076 \pm 0.0004$ | $295 \pm 20$ | $0.83 \pm 0.10$ |
| CFM01 | $26.7 \pm 2.0^{(24 h)}$ | $0.0018 \pm 0.0000$ | $413 \pm 12$ | $0.55 \pm 0.05$ |
| CFM1 | $876 \pm 99^{(114 h)}$ | $0.108 \pm 0.026$ | $70 \pm 2.3$ | $14.9 \pm 2.2$ |
| CFMt1 | $1210 \pm 75^{(114 h)}$ | $0.164 \pm 0.002$ | $70 \pm 7.9$ | $13.8 \pm 0.3$ |
| CMtS09 | $1540 \pm 75^{(48 h)}$ | $0.128 \pm 0.034$ | $337 \pm 90$ | $32.0 \pm 4.3$ |
| CMtS10 | $1370 \pm 40^{(72 h)}$ | $0.103 \pm 0.003$ | $254 \pm 54$ | $26.7 \pm 2.2$ |
| CMtS091 | $1890 \pm 110^{(48 h)}$ | $0.168 \pm 0.015$ | $327 \pm 30$ | $39.3 \pm 2.3$ |
| CMtS101 | $2000 \pm 79^{(72 h)}$ | $0.156 \pm 0.008$ | $226 \pm 8.5$ | $26.3 \pm 1.7$ |

$P_{max}$ the maximum amount of produced maleate. Data are presented as the mean $\pm$ standard deviation of three independent experiments ($n = 3$)

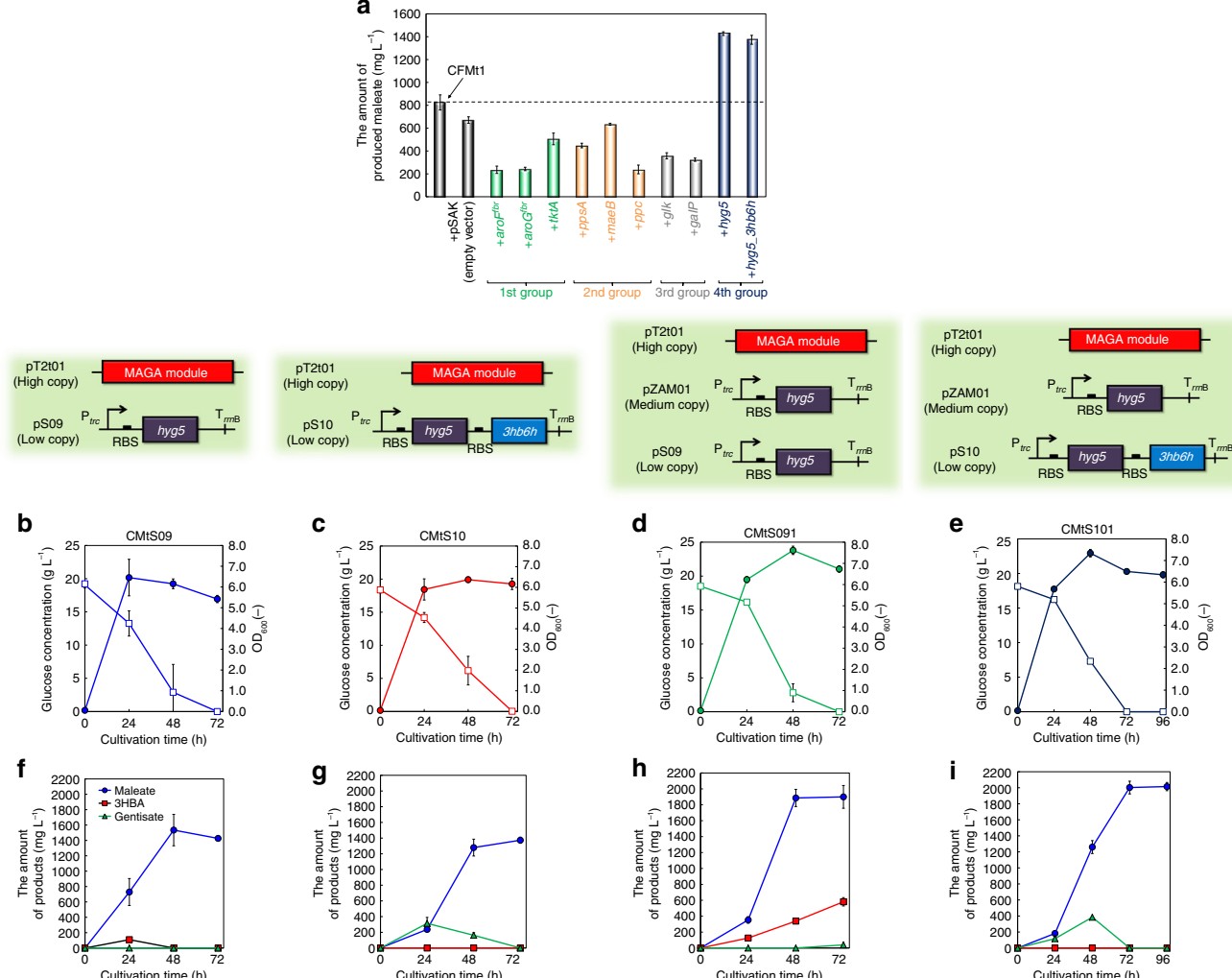

**Fig. 7** Improvement of rate limiting steps in the constructed pathway of maleate production. **a** The amount of maleate produced by each candidate transformant after 48 h of cultivation. Time courses of **b**–**e** the bacterial growth (filled symbols) and glucose concentration (open symbols), and **f**–**i** produced maleate (circles), 3HBA (squares), and gentisate (triangles) in the cultures of CMtS09, CMtS10, CMtS091 and CMtS101. Data are presented as the mean $\pm$ standard deviation of three independent experiments ($n = 3$)

are the parental strains of CFT5. ATCC 31882 and CFT3 carrying pT2t01 were named CFM00 and CFM01, respectively. Figure 6c–f shows the culture profiles of CFM00 and CFM01. The maximum levels of produced maleate were 41 and 27 mg L$^{-1}$, respectively (Fig. 6d, f). On the other hand, large amounts of L-phenylalanine and organic acids as by-products were produced in both cases (Fig. 6d, f and Supplementary Table 3).

**Improvement of a rate limiting step in maleate production**. To further improve the maleate production, we tried to identify and improve the rate limiting step in the synthetic pathway of maleate production, focusing on several genes, which were classified into four categories (Fig. 1). The first group included genes generally used to enhance the biosynthesis pathway of aromatic amino acids. Two 3-deoxy-D-heptulosonate-

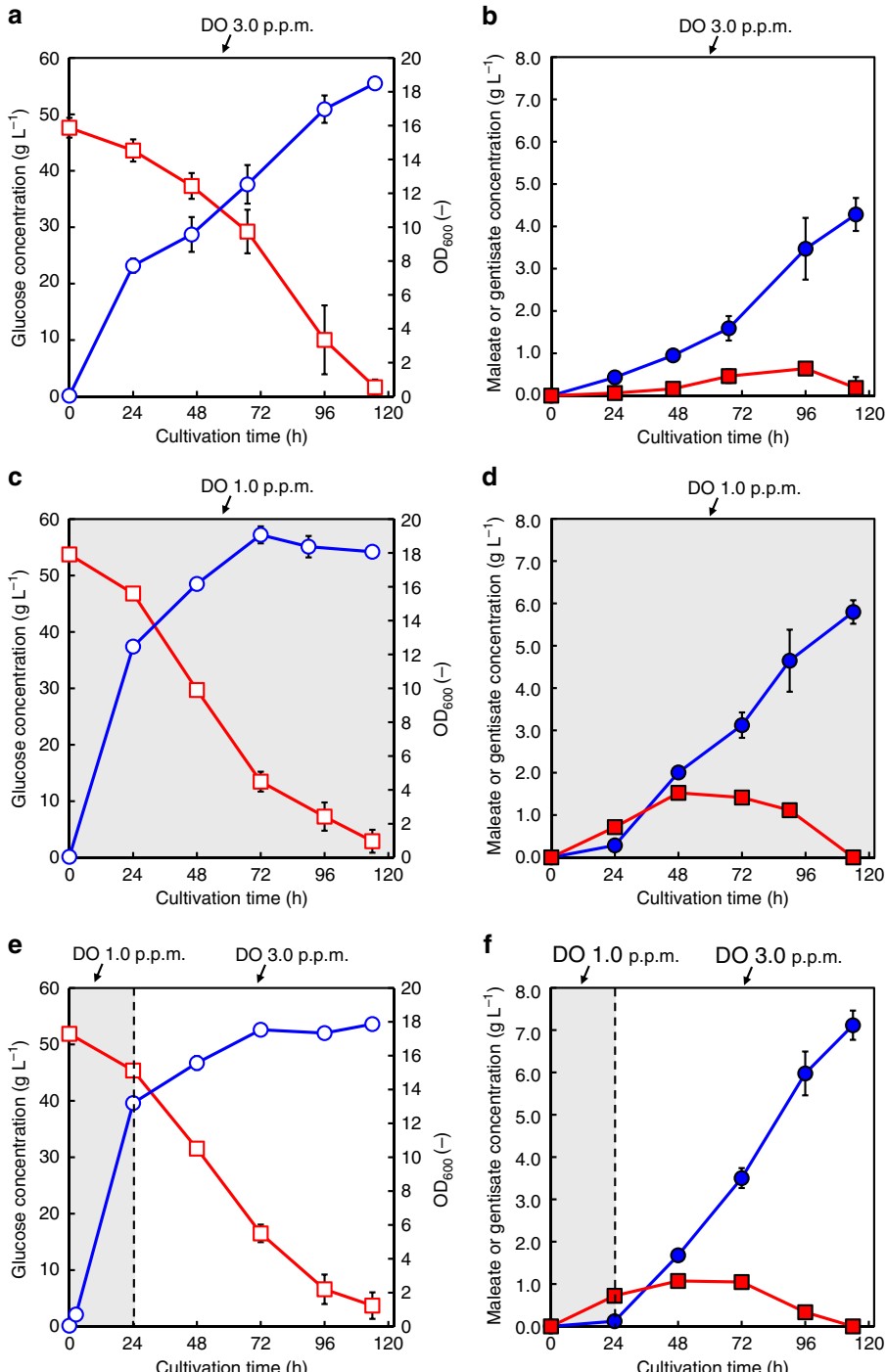

**Fig. 8** Batch culture of CMtS101 in a 1-L jar fermentor under three different DO conditions. Time courses of **a**, **c** and **e** the bacterial cell growth (circles) and glucose concentration (squares), and **b**, **d** and **f** the amount of produced maleate (circles) and gentisate (squares). Data are presented as the mean ± standard deviation of three independent experiments (n = 3)

7-phosphate synthases (AroF and AroG), which improve the condensation of erythrose-4-phosphate (E4P) and PEP and transketolase (TktA), which enhances the E4P availability, were used as the candidates[30–33]. In this study, we used modified AroF and AroG, whose feedback inhibition was deregulated (AroF[fbr] and AroG[fbr], respectively)[34, 35]. The second group included genes whose expression levels increased in the salicylate production by CFT5, as we have previously reported[27]. Phosphoenolpyruvate carboxylase (Ppc), malic enzyme B (MaeB) and phosphoenolpyruvate synthase (PpsA) were

selected as the candidates. The third group included genes involved in the glucose uptake system in CFT5. The genes encoding Glk and GalP were integrated into the genome of CFT5, whereas the glucose uptake PTS system was completely inactivated[27]. These two genes were also used as the candidates. The fourth group included genes needed to increase the carbon flux from chorismate to maleate, and *hyg5* and the GA module were adopted as the candidates. All these genes were individually cloned into pSAK, which is a low-copy-number vector, and introduced into CFMt1.

**Table 2 Summary of maleate production in the culture of CMtS101 using a 1-L jar fermentor**

| DO (p.p.m.) | $P_{max}^{(24 h)}$ (g L$^{-1}$) | Maleate yield (mol mol$^{-1}$) | Glucose uptake rate$^{(0-24 h)}$ (mg L$^{-1}$ h$^{-1}$) | Glucose uptake rate$^{(24-114 h)}$ (mg L$^{-1}$ h$^{-1}$) | Maleate production rate$^{(24-114 h)}$ (mg L$^{-1}$ h$^{-1}$) |
|---|---|---|---|---|---|
| 3.0 | $4.28 \pm 0.39$ | $0.139 \pm 0.01$ | $168 \pm 58$ | $437 \pm 18$ | $44.6 \pm 4.0$ |
| 1.0 | $5.80 \pm 0.28$ | $0.171 \pm 0.005$ | $289 \pm 77$ | $457 \pm 21$ | $60.4 \pm 2.9$ |
| 1.0–3.0 | $7.11 \pm 0.35$ | $0.221 \pm 0.004$ | $271 \pm 22$ | $434 \pm 26$ | $74.1 \pm 3.6$ |

DO dissolved oxygen, $P_{max}$ the maximum amount of produced maleate. Data are presented as the mean $\pm$ standard deviation of three independent experiments ($n = 3$)

Figure 7a shows the amount of produced maleate after 72 h of cultivation of each transformant. Using CFMtS09 and CFMtS10, carrying the genes categorised into the fourth group, i.e., those involved in the conversion of chorismate to maleate, the amounts of produced maleate were drastically increased compared to that produced by CFMt1, carrying the MAGA module with a single copy of each of *mps2*, *hbzF*, *hyg5* and *3hb6h*. However, the maleate production by transformants carrying the genes classified in the other groups decreased. Figure 7b, c, f, g shows the culture profiles of CFMtS09 and CFMtS10. The maximum levels of produced maleate were 1540 and 1370 mg L$^{-1}$ after 48 and 72 h of cultivation, respectively. In the case of CFMtS09, maleate and 3HBA were simultaneously produced until 24 h of cultivation, after which time the produced 3HBA was converted to maleate between 24 and 48 h of cultivation (Fig. 7f). In the case of CFMtS10, almost the same amounts of maleate and gentisate were produced until 24 h. After 24 h of cultivation, the produced gentisate was converted to maleate until 72 h of cultivation (Fig. 7g). Thus, we suggested one of rate limiting steps in maleate production by *E. coli* and successfully improved the process by optimising the synthetic pathway of maleate. Although 3HBA and gentisate formation was confirmed in the cultures of CFMtS09 and CFMtS10, these produced intermediates were completely converted to maleate after 48 or 72 h of cultivation, respectively (Fig. 7f, g).

To further improve the maleate production, we tried to enhance the carbon flux from chorismate to 3HBA by introducing pZAM01, which is a medium-copy-number vector carrying the *hyg5* gene under control of the *trc* promoter, into CFMtS09 and CFMtS10. Figure 7d, e, h, i shows the culture profiles of CMtS091 and CMtS101. The maximum amounts of produced maleate and the maleate yields from carbon sources were 1890 and 2000 mg L$^{-1}$ and 0.168 and 0.156 mol mol$^{-1}$ after 48 and 72 h of cultivation, respectively (Table 1). Unlike CFMtS09, 3HBA continued to accumulate in the culture supernatant of CMtS091, along with maleate formation (Fig. 7f, h). Although accumulation of gentisate in CMtS101 showed a similar behaviour to that in CFMtS10, the final concentration of produced maleate in the culture of CMtS101 improved compared to that in CFMtS10 (Fig. 7g, i). We also analysed the presence of maleylpyruvate, which is one of the intermediates in the maleate pathway (Fig. 1), in culture supernatants using high-performance liquid chromatography (HPLC). However, maleylpyruvate formation was not confirmed in any experiment evaluating the maleate production. A standard sample of maleylpyruvate was enzymatically synthesised from gentisate as described in Methods section. In addition, we performed the maleate production using [U-$^{13}$C]glucose as the carbon source to confirm the maleate production from glucose. Supplementary Fig. 6 shows the results. The specific peak of maleate–bis (trimethylsilyl) ester derivative derived from maleate using [$^{12}$C]glucose was composed of $m/z = 245$. However, that from maleate using [U-$^{13}$C]glucose was mainly consisted of $m/z = 249$ (over 98%). These results should strongly support that maleate was produced mainly from glucose.

**Maleate production using a 1-L jar fermentor**. A batch culture of CFTtS101 was performed using a 1-L jar fermentor. The initial concentration of glucose was ~50 g L$^{-1}$. Figure 8 shows the culture profiles of CFTtS101 in the jar fermentor. In this study, three different DO conditions were tested. First, the culture of CFTtS101 was performed with DO maintained at 3.0 p.p.m. during the entire cultivation. As shown in Fig. 8b, the amount of maleate produced by CFTtS101 reached ~4.3 g L$^{-1}$ after 114 h of cultivation. However, the bacterial growth rate decreased after 24 h of cultivation (Fig. 8a), and the maleate yield from carbon sources was 0.139 mol mol$^{-1}$ (Table 2). Then, the culture of CFTtS101 was performed with DO maintained at 1.0 p.p.m. during the entire cultivation. As shown in Fig. 8c, d, the initial bacterial growth improved, and the amount of produced maleate reached 5.8 g L$^{-1}$ after 114 h of cultivation. Finally, the culture of CFTtS101 was performed with DO shifted from 1.0 to 3.0 p.p.m. after 24 h of cultivation. As shown in Fig. 8e, f, the bacterial growth was maintained at the same level as in the case of DO maintained at 1.0 p.p.m. during the entire cultivation, but the maximum amount of produced maleate was 7.1 g L$^{-1}$, which was superior to that obtained with DO maintained at 3.0 p.p.m. during the entire cultivation. The maleate yield from carbon sources was 0.221 mol mol$^{-1}$, which was the highest value obtained in the present study (Table 2). Thus, the control of DO was shown to be one of significant factors in the production of maleate using *E. coli*.

## Discussion

Maleate is one of the most important chemicals, along with other industrially valuable four-carbon dicarboxylic acids, including succinate, malate and fumarate. A large number of reports have been published regarding the production of these dicarboxylic organic acids using various microbes as the host strains[13–16, 36, 37]. However, the microbial production of maleate has not been previously achieved, despite the tremendous global market and demand[11]. In the present study, we achieved the microbial production of maleate by using a metabolic engineering approach.

Maleate does not appear in metabolic pathways in most microorganisms, whereas succinate, malate and fumarate are usually produced in the TCA cycle[14–16]. Although there is a possibility to synthesise maleate from fumarate using maleate isomerase, which originally catalyses isomerisation of maleate to fumarate, the reverse reaction from fumarate to maleate has not been achieved in previous studies of maleate isomerase[38–40]. Thus, to produce maleate from simple carbon sources using *E. coli*, a novel synthetic pathway for maleate must be constructed. In the present study, we focused on two different pathways, the polyketide biosynthesis pathway and benzene ring cleavage pathway. The genes involved in the conversion of chorismate to maleate were screened and used to construct a maleate biosynthesis pathway in *E. coli* (Fig. 1). We have previously reported the production of 3HBA, which is a primary compound in the

maleate synthetic pathway, using *Streptomyces hygroscopicus* hyg5-expressing *E. coli*, in which pyruvate-forming reactions were inactivated[27]. Using this modified strain, maleate production from simple carbon sources, mainly glucose, was achieved in this study. After the synthetic pathway of maleate from glucose was constructed, the pathway converting chorismate to maleate was optimised. As an additional application, construction of a bypass to gentisate from waste aromatic compounds or lignin degradation products would enable the strains created in this study to produce maleate from various carbon sources, including waste materials or biomass derivatives.

In the production of aromatic compounds and their derivatives, the bottleneck mainly occurs in the shikimate pathway. There are numerous reports describing deregulation of the feedback inhibition, enhancement of the shikimate pathway, and increased accumulation of PEP and E4P, carried out to increase the production of target compounds[30–35]. AroG, AroF, TktA and PpsA are usually overexpressed to improve the productivity of host strains. We also tried to increase the amount of produced maleate in accordance with these classical strategies. However, overexpression of the enzymes resulted in a negative effect on maleate production by the derivatives of CFT5. This may be due to the fact that the carbon flux to chorismate was already optimised in CFT5, without introduction of these genes. Overexpression of the genes could cause an imbalance in the carbon flux or some transcriptional interference. We thus tried to fine-tune the constructed pathway of maleate from chorismate. Consequently, the maximum amount of produced maleate and the production rate drastically increased. Enhancement of the carbon flux from chorismate to 3HBA and gentisate (overexpression of hyg5 and 3hb6h) improved the maleate production and glucose uptake rate. In the test tube culture, CMtS091 (overexpressing two hyg5 genes in addition to the MAGA module) showed the highest yield and production rate among all transformants, which were 22.1-fold and 47.3-fold higher than those in CFM00, respectively. The maximum level of produced maleate was achieved in the culture of CMtS101 (overexpressing two hyg5 genes and one 3hb6h gene in addition to the MAGA module), which was 49-fold greater than that of CFM00. These results indicate that our platform strain CFT5 is versatile and can be used to synthesise valuable chemicals by extending the chorismate pathway. In this study, to increase the maleate production, three copies of the same gene involved in the production of 3HBA (hyg5) were introduced into three different vectors, and two copies of the same gene involved in the production of gentisate (3hb6h) were introduced into two different vectors. A total of seven genes were introduced into our base strain CFT5 using three vectors with different copy numbers in order to increase the carbon flux from glucose to maleate. The intracellular pool of chorismate is increased in CFT5 without additional introduction of genes enhancing the biosynthesis pathway of aromatic amino acids, whereas several genes, such as the feedback-deregulated aroF, aroG, aroB, aroL, tktA and ppsA, must be overexpressed when using general strategies designed to enhance this pathway[30, 31, 41]. Thus, our CFT5 strain, whose carbon flux to chorismate was increased without introduction of any genes, may have the advantage in creating strains carrying multiple genes involved in the production of aromatic compounds of interest.

Using a 1-L jar fermentor, a batch culture of CMtS101 (overexpressing two hyg5 genes and one 3hb6h gene in addition to the MAGA module) was performed. The concentration of maleate reached 7.1 g L$^{-1}$ after 114 h of cultivation under optimised DO conditions. The yield of maleate from the carbon sources was 0.221 mol mol$^{-1}$. Under optimised conditions, DO was shifted from 1.0 to 3.0 p.p.m. after 24 h of cultivation. In the synthetic pathway of maleate from chorismate, 2 mol of oxygen are essential to cleave the benzene ring and form maleylpyruvate and are considered one of the key factors. However, in the case of DO maintained at 3.0 p.p.m. throughout the cultivation, the initial bacterial growth until 24 h of cultivation was inferior to that observed under the optimised DO conditions (DO shifted from 1.0 to 3.0 p.p.m. after 24 h of cultivation) and under conditions of DO maintained at 1.0 p.p.m. during the entire culture. Although the accumulation of intermediates such as gentisate was not confirmed, the production of maleate also decreased compared to that obtained under the other two conditions (Fig. 8). The introduction of the synthetic pathway of maleate and the maintenance of oxygen stress at 3.0 p.p.m. could cause some kind of a metabolic burden[6]. We also performed culture with DO maintained at 1.0 p.p.m. during the entire cultivation. In this experiment, the maximum amount of the gentisate intermediate reached 1.5 g L$^{-1}$, whereas that obtained under the optimised DO conditions was 1.0 g L$^{-1}$. This may be attributed to the low DO level, which was not sufficient to cleave the benzene ring and produce maleate. The production of maleate under DO maintained at 1.0 p.p.m. during the entire culture was also lower than that obtained under the optimised DO conditions. This may be attributed to a low DO level in the production phase of maleate (after 24 h) in the case of DO maintained at 1.0 p.p.m. during the entire culture.

As additional applications of the benzene ring cleavage pathway, the production of adipate and its precursor, muconic acid (MA), has been widely researched[41–43]. In these studies, MA was obtained by the oxidation of catechol, which is a common intermediate of the pathway. Although chemical treatment is essential to produce adipate, which is one of the most valuable organic acids, along with maleate, microbial conversion of MA to adipate has recently been achieved using a novel enoate reductase[44]. Thus, the production of adipate directly from simple sugars may be performed in the near future. In the present study, we successfully produced industrially important maleate directly from simple carbon sources using microbial catalysis for the first time. The benzene ring cleavage pathway involves the oxidative reaction of the benzene ring to obtain normal dicarboxylic acids such as maleate and MA. Thus, the combination of the benzene ring cleavage pathway with the chorismate pathway may become one of the powerful strategies for the production of various dicarboxylic acids and their derivatives.

In conclusion, this study successfully demonstrated, for the first time, microbial production of industrially valuable maleate via the construction of a novel synthetic pathway to the best of our knowledge. After screening of a number of genes needed to construct the biosynthesis pathway, hyg5 from *S. hygroscopicus* ATCC 29253, 3hb6h from *R. jostii* RHA1, mps2 from *Rhodococcus* sp. strain NCIMB 12038, and hbzF from *P. alcaligenes* NCIMB 9867 were selected as the suitable genes. The carbon flux from chorismate to gentisate was a significant step in the pathway. In the constructed maleate synthetic pathway, 2 mol of pyruvate are released to produce 1 mol of maleate (Fig. 1). Recycling of pyruvate synthesised during maleate production would further increase the maleate production and yield. Large-scale and fed-batch fermentations may also be effective to increase the production of maleate. Our strategy combining the benzene ring cleavage pathway and chorismate pathway should be applicable to the production of other versatile dicarboxylic acid monomers or their derivatives from renewable feedstocks.

## Methods

**Strains and plasmid construction.** *E. coli* NovaBlue competent cells (Novagen, Cambridge, MA, USA) were used for gene cloning. Polymerase chain reaction (PCR) was performed using the KOD FX Neo DNA polymerase (Toyobo, Osaka,

Japan) and the appropriate primer pairs. Each gene was assembled with the respective plasmid using Gibson Assembly (New England Biolabs, Ipswich, MA, USA). Plasmids were transformed into bacterial strains using a Gene Pulser II (Bio-Rad, Hercules, CA, USA). Where applicable, 100 μg mL$^{-1}$ ampicillin, 50 μg mL$^{-1}$ kanamycin, and 15 μg mL$^{-1}$ chloramphenicol were added to media for selection.

The strains, plasmids and oligonucleotides used in this study are summarised in Supplementary Tables 4, 5. The accession numbers of proteins to construct a maleate synthesis pathway are summarised in Supplementary Table 6.

pTcgl3026, pT3hb6h and pTxlnD were constructed as follows. Synthetic genes corresponding to cgl3026 from Corynebacterium glutamicum, 3hb6h from Rhodococcus jostii RHA1, and xlnD from Pseudomonas alcaligenes NCIMB 9867, optimised for the Escherichia coli codon usage, were obtained from a commercial source (Invitrogen) (Supplementary Note 1). The 3hb6n and xlnD genes were amplified by PCR using the synthetic gene fragments as the templates with the primer pairs 3hb6h_f and 3hb6h_r and xlnD_f and xlnD_r, respectively. The amplified fragments were cloned into the KpnI site of pTrcBhyg5[27], and the resulting plasmids were designated pT3hb6h and pTxlnD, respectively. The synthetic gene fragment of cgl3026 was directly cloned into the KpnI site of pTrcBhyg5, and the resulting plasmid was designated pTcgl3026. pZA23hbzF, pZA23psal, pZA23ppu1, pZA23palc, pZA23rsp and pZA23ppu2 were constructed as follows. Synthetic genes corresponding to hbzF from P. alcaligenes NCIMB 9867, mps0 from Pseudaminobacter salicylatoxidans, sgp1 from Pseudomonas putida, mps1 from P. alcaligenes NCIMB 9867, mps2 from Rhodococcus sp. strain NCIMB 12038, and mps3 from P. putida, optimised for the E. coli codon usage, were obtained from a commercial source (Invitrogen) (Supplementary Note 1). First, the synthetic gene fragment of hbzF was directly cloned into the HindIII site of pZA23MCS, and the resulting plasmid was designated pZA23hbzF. Then, mps0, sgp1, mps1, mps2 and mps3 were individually cloned into the KpnI site of pZA23hbzF, and the resulting plasmids were designated pZA23psal, pZA23ppu1, pZA23palc, pZA23rsp and pZA23ppu2, respectively.

The cassette containing mps2 and hbzF under control of the P$_{AlacO1}$ promoter is referred to as the MA module, whereas that containing hyg5 and 3hb6h is referred to as the GA module (Supplementary Fig. 1a, b). pT2c101, pT2c102, pT2c103 and pT2c104 were constructed as follows. Each MA module was amplified by PCR using pZA23ppu1, pZA23palc, pZA23rsp or pZA23ppu2 as the template with the primer pair plac_to_ptrc_f and plac_to_ptrc_r. The plasmid fragment of pT3hb6h was also amplified by PCR using pT3hb6h as the template with the primer pair inv_ptrc_840_f and inv_ptrc_840_r. Each MA module was cloned into the amplified pT3hb6h, and the resulting plasmids were designated pT2c101, pT2c102, pT2c103 and pT2c104, respectively. Similarly, the plasmid fragment of pTcgl3026 was also amplified by PCR using pTcgl3026 as the template with the primer pair inv_ptrc_840_f and inv_ptrc_840_r. Each MA module was cloned into the amplified pTcgl3026, and the resulting plasmids were designated pT2c201, pT2c202, pT2c203 and pT2c204, respectively.

pT2c301 was constructed as follows. First, the gene set of mps2 and hbzF was amplified by PCR using pZA23rsp as the template with the primer pair mpr_hbz_ptrc_f and mpr_hbz_ptrc_r. The plasmid fragment of pTrcHisB was also amplified by PCR using pTrcHisB as the template with the primer pair inv_ptrc_f and inv_ptrc_r. The amplified gene set of mps2 and hbzF was cloned into the amplified pTrcHisB, and the resulting plasmid was designated pTmpsR. We further refer to the cassette containing mps2 and hbzF under control of the P$_{trc}$ promoter as the MAt module. Then, the MAt module was amplified by PCR using pTmpsR as the template with the primer pair ptrc_to_ptrc_840_f and ptrc_to_ptrc_840_r. The amplified fragment was cloned into the amplified pT3hb6h, and the resulting plasmid was designated pT2c301.

pT2t01 and pT2t02 were constructed as follows. First, the gene set of hyg5 and 3hb6h was amplified by PCR using pT3hb6h as the template with the primer pair ptrc_rbs_pz_psti_f and ptrc_rbs_pz_psti_r. The plasmid fragment of pZA23rsp was also amplified by PCR using pZA23rsp as the template with the primer pair inv_pZ_ptsi_f and inv_pZ_ptsi_r. The amplified gene set of hyg5 and 3hb6h was cloned into the amplified pZA23rsp, and the resulting plasmid was designated pZAt01. Then, the gene set of mps2, hbzF, hyg5 and 3hb6h was amplified by PCR using pZAt01 as the template with the primer pair mpr_to_ptrc_f and 3hb6h_to_ptrc_r. The amplified fragment was cloned into the amplified pTrcHisB, and the resulting plasmid was designated pT2t01. The cassette containing mps2, hbzF, hyg5 and 3hb6h under control of the P$_{trc}$ promoter is referred to as the MAGA module (Supplementary Fig. 1d). Additionally, the gene set of mps2 and hbzF was amplified by PCR using pZA23rsp as the template with the primer pair mpr_hbz_to_pt_hind_f and mpr_hbz_to_pt_hind_r. The amplified gene set of mps2 and hbzF was cloned into the HindIII site of pT3hb6h, and the resulting plasmid was designated pT2t02. The cassette containing hyg5, 3hb6h, mps2 and hbzF under control of the P$_{trc}$ promoter is referred to as the GAMA module (Supplementary Fig. 1d).

pSAK was constructed as follows. Fragments of the gene of chloramphenicol resistance, SC101 origin, and P$_{AlacO1}$ promoter were amplified by PCR using pZA33luc, pZS4Int-laci, and pZA23MCS as the templates with the primer pairs cmr_f and cmr_r, sc101_f and sc101_r, and palac_f and palac_r, respectively. The three fragments were circularised, and the resulting plasmid was designated pSAK.

pS01 and pS02 were constructed as follows. Synthetic genes corresponding to aroF$^{fbr}$ and aroG$^{fbr}$ from E. coli were obtained from a commercial source (Invitrogen) (Supplementary Note 1). The proline at position 148 in AroF was

substituted with leucine to generate L-tyrosine-insensitive AroF[34], and substitution of aspartate with asparagine at position 146 in AroG was performed to obtain L-phenylalanine-insensitive AroG[35]. The aroF$^{fbr}$ and aroG$^{fbr}$ genes were amplified by PCR using the synthetic gene fragments as the templates with the primer pairs arof_f and arof_r and arog_f and arog_r, respectively. The amplified fragments were cloned into the KpnI site of pSAK, and the resulting plasmids were designated pS01 and pS02.

pS03, pS04, pS05, pS06, pS07 and pS08 were constructed as follows. The tktA, ppc, maeB, ppsA, glk and galP genes were amplified by PCR using E. coli K-12 MG1655 genomic DNA as the template with the primer pairs tkta_f and tkta_r, ppc_f and ppc_r, maeb_f and maeb_r, ppsa_f and ppsa_r, glk_f and glk_r, and galp_f and galp_r, respectively. The amplified fragments were cloned into the KpnI site of pSAK, and the resulting plasmids were designated pS03, pS04, pS05, pS06, pS07 and pS08, respectively.

pS09 and pS10 were constructed as follows. In this study, the hyg5 gene under control of the P$_{trc}$ promoter was designated the Hyg5 cassette (Supplementary Fig. 1e). The Hyg5 cassette and GA module were amplified by PCR using pTrcBhyg5 and pT3hb6h as the templates, respectively, with the primer pair ptrc_psak_f and ptrc_psak_r. The amplified fragments were circularised with the fragments of the gene of chloramphenicol resistance and SC101 origin, and the resulting plasmids were designated pS09 and pS10, respectively (Supplementary Fig. 1e).

pA09 was constructed as follows. The Hyg5 cassette was amplified by PCR using pTrcBhyg5 as the template with the primer pair ptrc_pza23_f and ptrc_pza23_r. The plasmid fragment of pZA23MCS, except the promoter region, was also amplified by PCR using pZA23MCS as the template with the primer pair inv_pza23_f and inv_pza23_r. The Hyg5 cassette was cloned into the amplified pZA23MCS, and the resulting plasmid was designated pA09 (Supplementary Fig. 1f).

pETM1 was constructed as follows. The mps2 gene was amplified by PCR using the synthetic gene fragment of mps2 as the template with the primer pairs mps2_pet22_f and mps2_pet22_r. The amplified fragment was cloned into the NdeI and HindIII sites of pET-22b(+) (Novagen), and the resulting plasmids were designated pETM1.

All of the transformants constructed in this study are listed in Supplementary Table 4.

**Culture conditions**. M9Y medium was used for maleate production in 5-mL test tube-scale cultures. M9Y minimal medium contains (per litre): glucose, 20 g; yeast extract, 5 g; NaCl, 0.5 g; Na$_2$HPO$_4$·12H$_2$O, 17.1 g; KH$_2$PO$_4$, 3 g; NH$_4$Cl, 2 g; MgSO$_4$·7H$_2$O, 246 mg; CaCl$_2$·2H$_2$O, 14.7 mg; FeSO$_4$·7H$_2$O, 2.78 mg; thiamine hydrochloride, 10 mg; L-phenylalanine, 100 mg; L-tyrosine, 40 mg; and L-trypto-phan, 40 mg. (Phe, Tyr and Trp were included because CFT5 is auxotrophic for these amino acids.) For the culture of CFT5 derivative strains, 10 mM sodium pyruvate was added to the medium to encourage bacterial growth in the initial phase. When needed, ampicillin, kanamycin and/or chloramphenicol were added to the medium to a final concentration of 100, 50 and 15 μg mL$^{-1}$, respectively. Each preculture was seeded to 5 mL of M9Y medium in a test tube at an initial OD$_{600}$ of 0.05. Tube-scale cultures were incubated at 37 °C in a shaker at 180 r.p.m. IPTG (0.1 mM) was added to the culture medium at OD$_{600}$ of 0.5.

Batch-scale cultures were performed in a 1.0-L jar fermentor with a 400-mL working volume. M9Y medium supplemented with 1% yeast extract was used for maleate production at this scale. The culture medium (400 mL) in the jar fermentor was inoculated with 20 mL of a preculture. To maintain the pH at 7.0 during cultivation, NH$_4$OH was automatically added to the medium. DO was maintained at 1.0–3.0 p.p.m. by automatically controlling the agitation speed from 200 to 800 r. p.m. and supplementing with air (pre-warmed to 37 °C) when needed. In this experiment, 0.1 mM IPTG was added to the medium 2.5 h after the initiation of cultivation. For cultures supplemented with 10 mM sodium pyruvate, the amounts of consumed glucose and sodium pyruvate were used for yield estimation, which was calculated as Yield (mol mol$^{-1}$) = (produced compound, mol)/(consumed glucose + pyruvate, mol).

**Preparation of maleylpyruvate**. Expression and purification of gentisate 1,2-dioxygenase from Rhodococcus sp. strain NCIMB 12038 (Mps2) was performed as follows. pETM1 was introduced into the BL21 (DE3) pLysS strain of E. coli, and the resultant transformant was named BL21MP. This transformant was grown in Luria–Bertani medium to an optical density (OD$_{600}$) of 0.5 at 37 °C, and then the cells were incubated for an additional 30 min at 25 °C. Protein expression was induced by the addition of IPTG to a final concentration of 0.5 mM. After incu-bation for an additional 24 h at 25 °C, the cells were collected by centrifugation. The cell pellet was resuspended in 50 mM phosphate buffer containing 150 mM NaCl (pH 8.0) and lysed by sonication, followed by Mps2 purification using a TALON metal affinity resin (TaKaRa Bio, Shiga, Japan) according to the manu-facturer's protocol.

The enzymatic synthesis of maleylpyruvate from gentisate was conducted as follows. A reaction mixture (1 mL) containing 26 mM gentisate and 123 μM purified Mps2 in 50 mM Tris-HCl buffer (pH 7.5) was incubated for 2 h at 30 °C. The reaction mixture, as well as samples of culture supernatants, were analysed using HPLC. Supplementary Fig. 7 shows a time-course chromatogram of the

reaction mixture. The candidate peak of maleylpyruvate, found at ~5.1 min, increased as the peak of gentisate, found at ~10.8 min, decreased. To further confirm the maleylpyruvate formation, the reaction mixture was analysed using an LCMS-8040 system (Shimadzu). Peaks of dehydrogenated and decarboxylated products of maleylpyruvate were observed around $m/z$ of 185 and 140, respectively. Thus, maleylpyruvate was successfully synthesised from gentisate using Mps2.

**Analytical methods**. Cell growth was monitored by measuring $OD_{600}$ with a UVmini-1240 spectrophotometer (Shimadzu, Kyoto, Japan). The concentration of glucose in the culture supernatant was measured using the Glucose CII test (Wako, Kyoto, Japan) following the manufacturer's protocol.

**GC–MS and HPLC analysis**. GC–MS was performed using a GC–MS-QP2010 Ultra instrument (Shimadzu) equipped with a CP-Sil 8 CB-MS capillary column (30 m × 0.25 mm × 0.25 μm; Agilent). Helium was used as the carrier gas to maintain a flow rate of 2.1 mL min$^{-1}$. The injection volume was 1 μL with a split ratio of 1:10. The amounts of produced maleate, gentisate and 3HBA were quantified as follows. The oven temperature was initially held at 150 °C for 1 min, then raised to 240 °C at 12 °C min$^{-1}$, raised further to 300 °C at 120 °C min$^{-1}$, and finally held at 300 °C for 3 min. The total running time was 10 min. The other settings were as follows: interface temperature, 250 °C; ion-source temperature, 200 °C, and electron impact ionisation at 70 eV. Dried residues of maleate, gentisate and 3HBA were derivatised for 60 min at 80 °C in 50 μL of $N$-methyl-$N$-trimethylsilyltrifluoroacetamide and 20 μL of pyridine prior to the analysis[45, 46]. Sorbitol was used as the internal standard.

The maleate concentration was determined in a culture supernatant, which was separated from the culture broth by centrifugation at 21,880×$g$ for 20 min. The culture supernatant was analysed by HPLC (Shimadzu) using a 5C$_{18}$-PAQ column (Nacalai Tesque, Kyoto, Japan). The column was operated at 30 °C with a flow rate of 1.2 mL min$^{-1}$. A dual-solvent system was used, consisting of solvent A (50 mM phosphate buffer, pH 2.5) and solvent B (acetonitrile). The gradient was initiated at 100% A (0–3 min), gradually shifted to a 50:50 mixture of A and B (3–6 min), which was kept between 6 and 7 min, and subsequently shifted to 100% A (7–13 min). Product concentrations were determined using an ultraviolet absorbance detector (SPD-20AV; Shimadzu) at 210 nm.

Concentrations of organic acids were also determined in culture supernatants, which were separated from the culture broth by centrifugation at 21,880×$g$ for 20 min. The concentrations of organic acids as by-products were determined using an organic acid analysis system (Shimadzu) consisting of an HPLC instrument equipped with a Shim-pack SPR-H column. The column was operated at 48 °C with a flow rate of 0.8 mL min$^{-1}$. CDD-10A was used as the detector. $p$-Toluenesulphonic acid at a 5 mM concentration was used as the mobile phase, and 20 mM bis-Tris containing 5 mM $p$-toluenesulphonic acid and 100 μM ethylenediaminetetraacetic acid was mixed immediately before the detection to enhance the sensitivity.

**Data availability**. The data that support the findings of this study are available from the corresponding author upon reasonable request. The NCBI accession codes for the proteins used in this study are summarised in Supplementary Table 6.

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

## Acknowledgements

This work has been supported by the RIKEN Center for Sustainable Resource Science, Special Postdoctoral Researcher Program, and by the FY2015 Incentive Research Projects, and by JSPS KAKENHI Grant Number JP17K14870, Grant-in-Aid for Young Scientists (B). The authors would like to thank Dr. T. Tanaka (Kobe University) for HPLC analysis and Enago (www.enago.jp) for the English language review.

## Author contributions

S.N. designed and performed the experiments. Y.M. performed LC–MS analysis. S.O. performed the experiments. S.N. wrote the paper. S.N. and T.S. revised the paper. A.K. supervised the research.
