## [Peer Review File · Nature Communications]

Reviewers' comments:

Reviewer #1 (Remarks to the Author):

Noda et al. report a novel pathway towards maleate. The novelty is the combination of terpenoid

biosynthesis pathway and the benzene ring cleavage pathway. Maleic acid anhydride is an important precursor for polymers and pharmaceuticals with a considerable market volume (1 million tons per annum). In Nature, maleate is an intermediate of aromatic compound degradation with gentisate as its immediate precursor. However, natural pathways from sugars or renewables do not lead to gentisate. The authors identified terpenoids biosynthesis as a natural pathway leading to compounds that can be converted to gentisate. The missing link was closed by generating gentisate from chorismate.

The experiments have been performed convincingly, all conclusions are supported by experimental evidence. Given the large market volume and, thus, the excellent application potential of this maleate process, this reviewer would suggest demonstration of the feasibility at a larger scale (not just 1L jar scale) and the demonstration that maleate production from alternative carbon sources is possible.

Reviewer #2 (Remarks to the Author):

The manuscript of Noda et al. reports the construction of an synthetic metabolic pathway for production of maleate in *E. coli*. The synthetic pathway and the achieved titers are clearly an important achievement. However, I have some points regarding the analytical methods and the interpretation of data in the final section. Another concern is the structure and the clarity of the manuscript. My detailed comments are listed below:

1. Validation of the analytical method and data on by-products

Fumarate is an isomer of maleate, and the authors must show that their analytical method is sufficiently selective to distinguish these compounds. This is especially important since fumarate is a potential byproduct in the benzene ring cleavage pathway. In general, to demonstrate proper functioning of the synthetic pathway it is important to show that that little byproducts or intermediates are produced, and most importantly quantify the major by-product pyruvate. The GC-MS method used by the authors should be able to detect maleylpyruvate and chorismate as well. On p 16 the authors state that this was tested with HPLC, why not GC-MS. The same on p. 11, "...chorismate pool increased.", how was chorismate measured.

2. Production of maleate from glucose

This section is not very clear, how can the authors be sure that maleate is produced from glucose and not from other precursors in the complex medium (I understand they added yeast extract). To show production from glucose one can simply feed ¹³-C glucose and demonstrate by GC-MS that maleate gets fully labeled. An even stronger point would be production with glucose as the sole carbon source. Further it would be helpful to see production and growth rates, check if they are correlated and eventually perform a carbon balance. The effect of DO is interesting but again there is very little information why the setpoints of 3 and 1 ppm were selected and how this affected

respiration of cells (e.g. providing rates of oxygen consumption).

3. Clear presentation of the plasmids, strains and genetic modifications.

The main contribution of the paper is the novel synthetic pathway shown in Figure 1. I encourage the authors to present this in a much clearer way. It is not clear to me from this Figure which modifications and strains were used. Ideally this information is included in Figure 1, e.g. it is unclear what 1st to 4th group refers to. The next problem is the many abbreviations in the text, what makes it often difficult to understand without going back to the supplement table 4. The authors should provide this information in the text, ideally graphically (maybe instead of the many bar plots).

Minor points

- P.15 "...we identified a bottleneck..." is misleading in this context. The authors see higher production when increasing the first step in their synthetic pathway, which can have several reasons. To demonstrate a genuine bottleneck in the pathway one has to measure intracellular metabolites and show e.g. increased chorismate, which decreases upon relief of the bottleneck.
- Yields should be consistent either in mol per mol or g per g
- The authors reported the production of maleate to be fermentative. This should be changed to aerobic batch cultivation.
- The time axes in Fig 3 a and b are not consistent: 72h vs. 24h. These were samples from the same experiment?
- The authors should clarify better the number of replicates in the experiments and how the data were statistically treated.
- Can the authors exclude that intracellular maleate is released during the sample preparation? They use very high g-forces and long centrifugation times: "The maleate concentration was determined in a culture supernatant, which was separated from the culture broth by centrifugation at $21,880 \times g$ for 20 min"
- Is there a reference for the statement on p 5: "however, microorganisms capable of degrading aromatic compounds do not have the ability to endogenously produce gentisate from sugars or other renewable carbon sources via the central metabolic microbial pathways such as the glycolysis pathway or pentose phosphate pathway".

Responses to the comments by reviewer 1

Thank you for your thoughtful comments that have helped us to improve our manuscript considerably. We revised our manuscript according your suggestion as follows.

Comments 1

Noda et al. report a novel pathway towards maleate. The novelty is the combination of terpenoid biosynthesis pathway and the benzene ring cleavage pathway. Maleic acid anhydride is an important precursor for polymers and pharmaceuticals with a considerable market volume (1 million tons per annum). In Nature, maleate is an intermediate of aromatic compound degradation with gentisate as its immediate precursor. However, natural pathways from sugars or renewables do not lead to gentisate. The authors identified terpenoids biosynthesis as a natural pathway leading to compounds that can be converted to gentisate. The missing link was closed by generating gentisate from chorismate. The experiments have been performed convincingly, all conclusions are supported by experimental evidence. Given the large market volume and, thus, the excellent application potential of this maleate process, this reviewer would suggest demonstration of the feasibility at a larger scale (not just 1L jar scale) and the demonstration that maleate production from alternative carbon sources is possible.

Response

As you pointed out, the culture using a larger jar fermentor is so significant in aspect of industrial application of our developed process. In order to perform the production of bulk chemicals using microbes at “the process development scale”, we have to carry out the culture using over 100 liter jar fermentor, whereas 1-10 liter of that is usually adopted at “the laboratory scale process^{1,2}”. For the process development scale, the expensive laboratory systems such as a larger scale jar fermentor have to be prepared. In the present study, we constructed the novel biosynthesis pathway of maleate and evaluated using the batch culture at a laboratory scale. The optimisation of culture at the process development scale such as fed-batch culture using larger jar fermentor will be performed for industrial application of this process in near future.

As you pointed out, the production of maleate from alternative carbon sources is one of impressive suggestions to demonstrate the versatility and utility of our process. According to your suggestion, we here performed the production of maleate from xylose, which is the main component of hemicellulosic biomass such as xylan. **Figure for revise 1** shows the result. Each parameter, the cell growth rate, uptake rate of substrates or maleate production was lower than that of using glucose as the carbon source (**Figure 7e, i**). In *E. coli* transformant used in this study, the metabolic pathway

was optimised for maleate production using glucose as the carbon source (e.g. PTS system was replaced to the combined system of galactose permease and glucokinase). In order to encourage maleate production using xylose, the metabolic pathway should be redesigned as suitable for metabolism of xylose. This significant topic will be undertaken in our future work.

Figure for revise 1 | Culture profile of CMtS101 during maleate production using 2% xylose as the carbon source. Time courses of (a) the glucose concentration (open symbols) and OD₆₀₀ (filled symbols), and (b) the amount of produced maleate. Data are presented as the mean ± standard deviation of three independent experiments.

References for revise

1. Margeot, A., Hahn-Hagerdal, B., Edlund, M., Slade, R. & Monot, F. New improvements for lignocellulosic ethanol. *Curr. Opin. Biotechnol.* **20**, 372–380 (2009). doi: 10.1016/j.copbio.2009.05.009.
2. Park, S.H., Kim, H.U., Kim, T.Y., Park, J.S., Kim, S.S. & Lee, S.Y. Metabolic engineering of *Corynebacterium glutamicum* for L-arginine production. *Nat. Commun.* **5**, 4618–4628 (2014). doi: 10.1038/ncomms5618.

Responses to the comments by reviewer 2

Thank you for your thoughtful comments that have helped us to improve our manuscript considerably. We revised our manuscript according your suggestion as follows.

Comments 1

Validation of the analytical method and data on by-products

Fumarate is an isomer of maleate, and the authors must show that their analytical method is sufficiently selective to distinguish these compounds. This is especially important since fumarate is a potential byproduct in the benzene ring cleavage pathway. In general, to demonstrate proper functioning of the synthetic pathway it is important to show that that little byproducts or intermediates are produced, and most importantly quantify the major by-product pyruvate. The GC-MS method used by the authors should be able to detect maleylpyruvate and chorismate as well. On p 16 the authors state that this was tested with HPLC, why not GC-MS. The same on p. 11, “..chorismate pool increased.”, how was chorismate measured.

Response

As you pointed out, fumarate is a potential byproduct in the benzene ring cleavage pathway. However, to obtain fumarate from this pathway, the enzymes converting maleylpyruvate to fumarate must be expressed in *E. coli*. In this pathway, after maleylpyruvate is converted to fumarylpyruvate by maleylpyruvate isomerase, fumarate is obtained from the precursor by 3-fumarylpyruvate hydrolase (**Figure for revise 1**). In addition, the detectable level of fumarate produced in TCA cycle was not also confirmed. About pyruvate formation, we reported in **Supplementary Table 3**.

In this work, we analysed a large number of samples in the process of constructing and optimising maleate producing *E. coli*, except for the presented samples in this paper. To perform high-throughput and cost-effective analysis, we adopted HPLC analysis to estimate the amount of produced maleate. In our HPLC method, maleate and fumarate can be completely separated (**Figure for revise 2**). Thus, the estimation of maleate using HPLC should be successfully demonstrated.

Actually, we did not estimate the amount of chorismate accumulated in *E. coli*. Our base strain CFT5 was designed to accumulate phosphoenol pyruvate (PEP) by inactivating the enzymes concerning PEP to pyruvate. PEP accumulation is one of key points to enhance the production of aromatic chemicals in microbe. In addition, the pathway to aromatic amino acids from chorismate was also completely inactivated in

CFT5. So, we assumed that the intracellular chorismate pool was expanded in CFT5. In our previous work, we reported the production of the chorismate derivatives using CFT5¹. 8 chemicals converted from chorismate via one or two step reactions were individually produced with the yield of over 0.2 mol mol⁻¹. These results strongly support that the chorismate pool in CFT5 is increased. However, as you pointed out, we did not measure the amount of produced chorismate (to be more precise, the carbon flux to chorismate would be increased). To avoid misleading to reader, we revise the manuscript.

*These results indicated that the combination of *hyg5*, *3hb6h*, *gdrs*, and *hbzF* would be suitable for maleate production using our base strain CFT5, whose carbon flux to chorismate is increased. (See page 11, line 182-186)*

Thus, our CFT5 strain, whose carbon flux to chorismate was increased without introduction of any genes, may have the advantage in creating strains carrying multiple genes involved in the production of aromatic compounds of interest. (See page 21, line 354-356)

Figure for revise 1 | The degrading pathway of gentisate to maleate or fumarate.

Figure for revise 2 | HPLC chromatogram of the mixture of maleate and fumarate standard samples.

Comments 2

Production of maleate from glucose

This section is not very clear, how can the authors be sure that maleate is produced from glucose and not from other precursors in the complex medium (I understand they added yeast extract). To show production from glucose one can simply feed ^{13}C glucose and demonstrate by GC-MS that maleate gets fully labeled. An even stronger point would be production with glucose as the sole carbon source. Further it would be helpful to see production and growth rates, check if they are correlated and eventually perform a carbon balance. The effect of DO is interesting but again there is very little information why the setpoints of 3 and 1 ppm were selected and how this affected respiration of cells (e.g. providing rates of oxygen consumption).

Response

As you pointed out, it is important to be sure that maleate is produced from glucose. So, we carried out the production of maleate using M9Y medium without additional glucose as well as with glucose. As shown in **Supplementary Figure 6 in the submitted manuscript**, the amount of produced maleate using M9Y without glucose as the carbon source was below $70 \text{ mg}\cdot\text{L}^{-1}$, whereas that using M9Y medium with 2% glucose was $1,210 \text{ mg}\cdot\text{L}^{-1}$ (**Fig. 6**). These results strongly support that maleate was produced mainly from glucose.

According to your suggestion, we also performed the maleate production using $[\text{U}-^{13}\text{C}]$ glucose as the carbon source, followed by GC-MS analysis. **Figure for revise 3** shows the result. The specific peak of maleate–bis(trimethylsilyl) ester derivative derived from maleate using $[\text{C}^{12}]$ glucose was composed of $m/z = 245$. However, that from maleate using $[\text{U}-^{13}\text{C}]$ glucose was mainly consisted of $m/z = 249$ (over 98%) (consider that the purity of $[\text{U}-^{13}\text{C}]$ glucose is about 99%). These results should strongly support that maleate was produced mainly from glucose in our work. We included these results into our manuscript as supplementary information.

In addition, we performed the maleate production using $[\text{U}-^{13}\text{C}]$ glucose as the carbon source to confirm the maleate production from glucose. Supplementary Figure 6 shows the results. The specific peak of maleate–bis(trimethylsilyl) ester derivative derived from maleate using $[\text{C}^{12}]$ glucose was composed of $m/z = 245$. However, that from maleate using $[\text{U}-^{13}\text{C}]$ glucose was mainly consisted of $m/z = 249$ (over 98%). These results should strongly support that maleate was produced mainly from glucose. (See page 16, line 270-276)

Supplementary Figure 6 | Labeling pattern of maleate produced by CMtS101 cultured using [¹²C]glucose or [U-¹³C]glucose as the carbon source. (a) Structure of maleate-bis(trimethylsilyl) ester derivative and m/z of the demethylated product. (b) Absolute intensity of m/z = 245 and 249 in the case of using [¹²C]glucose. (c) Absolute and (d) relative intensity of m/z = 245 and 249 in the case of using [U-¹³C]glucose. Data are presented as the mean ± standard deviation of three independent experiments. (See page 27, line 507-514 in supplementary information)

There are a lot of reports about microbial production of useful chemicals in aerobic condition. In some reports among them, dissolved oxygen (DO) of about 3.0 ppm (DO of 3.0 ppm at 37 °C is 0.43 v·v⁻¹) is adopted as aerobic condition²⁻⁴. In

addition, 2 mol of oxygen are essential to cleave the benzene ring and form maleylpyruvate and are considered one of the key factors in the synthetic pathway of maleate from chorismate (**Figure for revise 1**). According to those reports and the designed pathway requiring 2 mol of oxygen, we firstly performed the cultivation using 1-liter jar fermentor with DO maintained at 3.0 ppm. However, as shown in **Fig. 8a, b**, the initial cell growth and production rate of maleate was decreased after 24 h cultivation. In order to avoid the inhibition of cell growth and supply enough amount of oxygen to convert chorismate to maleate, we performed culture with DO shifted from 1.0 to 3.0 ppm after 24 h of cultivation. Although we also carried out the culture with DO maintained at 1.0 ppm, the maximum amount of the gentisate intermediate reached $1.5 \text{ g}\cdot\text{L}^{-1}$, whereas that obtained under the optimised DO conditions was $1.0 \text{ g}\cdot\text{L}^{-1}$. This may be attributed to the low DO level, which was not sufficient to cleave the benzene ring and produce maleate. Indeed, the amount of produced maleate in the optimised DO conditions (DO shifted from 1.0 to 3.0 ppm after 24 h of cultivation) was $7.1 \text{ g}\cdot\text{L}^{-1}$, which was superior to that in culture with DO maintained at 1.0 ppm throughout the cultivation ($5.8 \text{ g}\cdot\text{L}^{-1}$).

Figure for revise 3 | Labeling pattern of maleate produced by CMtS101 cultured using [¹²C]glucose or [U-¹³C]glucose as the carbon source. (a) Structure of maleate–bis(trimethylsilyl) ester derivative and m/z of the demethylated product. (b) Absolute intensity of m/z = 245 and 249 in the case of using [¹²C]glucose. (c) Absolute and (d) relative intensity of m/z = 245 and 249 in the case of using [U-¹³C]glucose.

Comments 3

Clear presentation of the plasmids, strains and genetic modifications.

The main contribution of the paper is the novel synthetic pathway shown in Figure 1. I encourage the authors to present this in a much clearer way. It is not clear to me from this Figure which modifications and strains were used. Ideally this information is included in Figure 1, e.g. it is unclear what 1st to 4th group refers to. The next problem is the many abbreviations in the text, what makes it often difficult to understand without going back to the supplement table 4. The authors should provide this information in the text, ideally graphically (maybe instead of the many bar plots).

Response

In this journal, the main text should be written in no more than 5,000 words. It may be difficult to explain the abbreviations in each section. However, clear presentation as you pointed out is one of significant points for the better comprehension to reader. According to your suggestion, we included information about our constructed strain in each figure.

Figure 1.

(See page 40, line 618-619)

Figure 2.

(See page 41, line 620-621)

Figure 3.

(See page 42, line 623-625)

Figure 5.

(See page 44, line 629-630)

Figure 6.

(See page 45, line 631-632)

Supplementary Figure 3 | Culture profile of CFM9 during maleate production. Time courses of (a) the glucose concentration (open symbols) and OD₆₀₀ (filled symbols), and (b) the amount of produced maleate. Data are presented as the mean \pm standard deviation of three independent experiments. (See page 24, line 487-493 in supplementary information)

Supplementary Figure 4 | Culture profile of CFMt2 during maleate production. Time courses of (a) the glucose concentration (open symbols) and OD₆₀₀ (filled symbols), and (b) the amounts of produced maleate and gentisate. Data are presented as the mean \pm standard deviation of three independent experiments. (See page 25, line 495-500 in supplementary information)

Minor points

Comments 4

P.15 "...we identified a bottleneck..." is misleading in this context. The authors see higher production when increasing the first step in their synthetic pathway, which can have several reasons. To demonstrate a genuine bottleneck in the pathway one has to measure intracellular metabolites and show e.g. increased chorismate, which decreases upon relief of the bottleneck.

Response

As you pointed out, our expression "...we identified a bottleneck..." and others may be misleading to readers. According to your suggestion, we revised the manuscript.

The metabolic engineering approach used to fine-tune the synthetic pathway drastically improved the maleate production and demonstrated that one of rate limiting steps existed in the conversion of chorismate to gentisate. In a batch culture of the optimised transformant, grown in a 1-L jar fermentor, the amount of produced maleate reached $7.1 \text{ g}\cdot\text{L}^{-1}$, and the yield was $0.221 \text{ mol}\cdot\text{mol}^{-1}$. (See page 2, line 18-22)

***Improvement of a rate limiting step in the production of maleate by E. coli.** To further improve the maleate production, we tried to identify and improve the rate limiting step in the synthetic pathway of maleate production, focusing on several genes, which were classified into four categories (Fig. 1). (See page 13, line 220-223)*

Thus, we suggested one of rate limiting steps in maleate production by E. coli and successfully improved the process by optimising the synthetic pathway of maleate. (See page 15, line 250-252)

***Figure 7 | Improvement of rate limiting steps in the constructed pathway of maleate production.** (a) The amount of maleate produced by each candidate transformant after 48 h of cultivation. Time courses of (b–e) the bacterial growth (filled symbols) and glucose concentration (open symbols), and (f–i) produced maleate (circles), 3HBA (squares), and gentisate (triangles) in the cultures of CMtS09, CMtS10, CMtS091, and CMtS101. Data are presented as the mean \pm standard deviation of three independent experiments. (See page 36, line 600-605)*

Comments 5

Yields should be consistent either in mol per mol or g per g

Response

According to your suggestion, we revised the manuscript.

The metabolic engineering approach used to fine-tune the synthetic pathway drastically improved the maleate production and demonstrated that one of rate limiting steps existed in the conversion of chorismate to gentisate. In a batch culture of the optimised transformant, grown in a 1-L jar fermentor, the amount of produced maleate reached $7.1 \text{ g}\cdot\text{L}^{-1}$, and the yield was $0.221 \text{ mol}\cdot\text{mol}^{-1}$. (See page 2, line 18-22)

By optimising batch culture conditions in a 1-L jar fermentor through the alteration of the amount of dissolved oxygen (DO), we successfully produced $7.1 \text{ g}\cdot\text{L}^{-1}$ maleate, which represented a $0.221 \text{ mol}\cdot\text{mol}^{-1}$ yield relative to the carbon sources added to the medium. (See page 6, line 82-85)

*The highest yield of gentisate was $0.165 \text{ mol}\cdot\text{mol}^{-1}$ in the culture of CFG1, carrying the *hyg5* and *cgl3026* gene set, while the yield in the culture of CFG2, carrying the *hyg5* and *3hb6b* gene set, was $0.130 \text{ mol}\cdot\text{mol}^{-1}$ (Supplementary Table 1). (See page 7-8, line 117-119)*

The highest yield of maleate produced from carbon sources was $0.189 \text{ mol}\cdot\text{mol}^{-1}$ in the culture of CFM1 after 48 h of cultivation. (See page 7-8, line 115-117)

The yield of maleate produced from carbon sources during the entire cultivation was $0.164 \text{ mol}\cdot\text{mol}^{-1}$ (Table 1). (See page 13, line 207-209)

The maximum amounts of produced maleate and the maleate yields from carbon sources were $1,890$ and $2,000 \text{ mg}\cdot\text{L}^{-1}$ and 0.168 and $0.156 \text{ mol}\cdot\text{mol}^{-1}$ after 48 and 72 h of cultivation, respectively (Table 1). (See page 15-16, line 259-261)

However, the bacterial growth rate decreased after 24 h of cultivation (Fig. 8a), and the maleate yield from carbon sources was 0.139 mol·mol⁻¹ (Table 2). (See page 17, line 284-285)

The maleate yield from carbon sources was 0.221 mol·mol⁻¹, which was the highest value obtained in the present study (Table 2). (See page 17, line 293-294)

The yield of maleate from the carbon sources was 0.221 mol·mol⁻¹. (See page 21, line 359-360).

For cultures supplemented with 10 mM sodium pyruvate, the amounts of consumed glucose and sodium pyruvate were used for yield estimation, which was calculated as Yield (mol·mol⁻¹) = [produced compound, mol]/[consumed glucose + pyruvate, mol]. (See page 6, line 139-141 in supplementary information)

Table 1. Summary of maleate production by each engineered *E. coli* strain.

Strain	P_{max} (mg·L⁻¹)	Maleate yield (mol·mol⁻¹)	Glucose uptake rate^(0 to 48 h) (mg·L⁻¹·h⁻¹)	Maleate production rate^(0 to 48 h) (mg·L⁻¹·h⁻¹)
CFM00	40.8 ± 4.3 ^(24 h)	0.0076 ± 0.0004	295 ± 20	0.83 ± 0.10
CFM01	26.7 ± 2.0 ^(24 h)	0.0018 ± 0.0000	413 ± 12	0.55 ± 0.05
CFM1	876 ± 99 ^(114 h)	0.108 ± 0.026	70 ± 2.3	14.9 ± 2.2
CFMt1	1,210 ± 75 ^(114 h)	0.164 ± 0.002	70 ± 7.9	13.8 ± 0.3

CMtS09	$1,540 \pm 75^{(48 \text{ h})}$	0.128 ± 0.034	337 ± 90	32.0 ± 4.3
CMtS10	$1,370 \pm 40^{(72 \text{ h})}$	0.103 ± 0.003	254 ± 54	26.7 ± 2.2
CMtS091	$1,890 \pm 110^{(48 \text{ h})}$	0.168 ± 0.015	327 ± 30	39.3 ± 2.3
CMtS101	$2,000 \pm 79^{(72 \text{ h})}$	0.156 ± 0.008	226 ± 8.5	26.3 ± 1.7

P_{\max} , the maximum amount of produced maleate.

(See page 37-38, line 612-613)

Table 2. Summary of maleate production in the culture of CMtS101 using a 1-L jar fermentor.

DO, dissolved oxygen; P_{\max} , the maximum amount of produced maleate.

DO (ppm)	$P_{\max}^{(24\text{ h})}$ ($\text{g}\cdot\text{L}^{-1}$)	Maleate yield ($\text{mol}\cdot\text{mol}^{-1}$)	Glucose uptake rate^(0 to 24 h) ($\text{mg}\cdot\text{L}^{-1}\cdot\text{h}^{-1}$)	Glucose uptake rate^(24 to 114 h) ($\text{mg}\cdot\text{L}^{-1}\cdot\text{h}^{-1}$)	Maleate production rate^(24 to 114 h) ($\text{mg}\cdot\text{L}^{-1}\cdot\text{h}^{-1}$)
3.0	4.28 ± 0.39	0.139 ± 0.01	168 ± 58	437 ± 18	44.6 ± 4.0
1.0	5.80 ± 0.28	0.171 ± 0.005	289 ± 77	457 ± 21	60.4 ± 2.9
1.0 to 3.0	7.11 ± 0.35	0.221 ± 0.004	271 ± 22	434 ± 26	74.1 ± 3.6

(See page 39, line 615-616)

Supplementary Table 1. Summary of the amounts of consumed glucose and yields of produced gentisate in cultures of CFG1, CFG2, and CFG3.

	Strains		
	CFG1	CFG2	CFG3
Consumed glucose ($\text{g}\cdot\text{L}^{-1}$)	5.7 ± 0.8	5.9 ± 0.5	17.9 ± 0.6
Gentisate yield ($\text{mol}\cdot\text{mol}^{-1}$)	0.165 ± 0.022	0.130 ± 0.002	0.022 ± 0.007

(See page 29, line 521-526 in supplementary information)

Supplementary Table 2. Summary of the amounts of consumed glucose and yields of produced maleate in eight transformants.

	Strains							
	CFM1	CFM2	CFM3	CFM4	CFM5	CFM6	CFM7	CFM8
Consumed glucose (g·L ⁻¹)	3.4 ± 0.1	4.8 ± 1.3	7.8 ± 0.6	6.0 ± 0.1	2.8 ± 0.1	1.2 ± 0.2	0.9 ± 0.2	1.1 ± 0.1
Maleate yield (mol·mol ⁻¹)	0.189 ± 0.018	0.477 ± 0.0093	0.0868 ± 0.0085	0.0842 ± 0.0086	0.117 ± 0.010	0.0896 ± 0.0401	0.0749 ± 0.0014	0.0528 ± 0.0023

(See page 30, line 528-529 in supplementary information)

Comments 6

The authors reported the production of maleate to be fermentative. This should be changed to aerobic batch cultivation.

Response

According to your suggestion, we revised the manuscript.

However, the microbial production of maleate has not been previously achieved, despite the tremendous global market and demand¹¹. (See page 18, line 301-303)

Comments 7

The time axes in Fig 3 a and b are not consistent: 72h vs. 24h. These were samples from the same experiment?

Response

The samples used for **Figure 3a**, b were obtained from the same experiment. In this experiment, gentisate was added to the medium after 4 h cultivation. The samples were collected at 4, 8, 22, and 72 h cultivation. The concentration of maleate was measured at 4, 8, and 22 h.

According your suggestion, we revised the manuscript.

*In the present study, we screened several genes encoding GDO, while hbzF from *P. alcaligenes* NCIMB 9867 was adopted as the gene encoding MPH^{28,29}. Using M9Y medium supplemented with gentisate (500–600 mg·L⁻¹), the gentisate-degrading ability of each strain (CFMP1–5) was evaluated. In this experiment, the additional gentisate was added to the medium after 4 h cultivation. (See page 8, line 127-131)*

Figure 3.

(See page 42, line 623-625)

Comments 8

The authors should clarify better the number of replicates in the experiments and how the data were statistically treated.

Response

According to your suggestion, we revised the manuscript.

Figure 7 | Improvement of rate limiting steps in the constructed pathway of maleate production. (a) The amount of maleate produced by each candidate transformant after 48 h of cultivation. Time courses of (b–e) the bacterial growth (filled symbols) and glucose concentration (open symbols), and (f–i) produced maleate (circles), 3HBA (squares), and gentisate (triangles) in the cultures of CMtS09, CMtS10, CMtS091, and CMtS101. Data are presented as the mean ± standard deviation of three independent experiments. (See page 36, line 600-605)

Comments 9

Can the authors exclude that intracellular maleate is released during the sample preparation? They use very high g-forces and long centrifugation times: “The maleate concentration was determined in a culture supernatant, which was separated from the culture broth by centrifugation at 21,880 × g for 20 min”

Response

As you pointed out, we here investigated whether high gravity force and long centrifuge time had affected the concentration of maleate or not. 4 different separation conditions was adopted and carried out. The culture supernatant of CMtS101 after 72 h was separated from the culture broth by centrifugation at $21,880 \times g$ for 20 min, $21,880 \times g$ for 5 min, $10,000 \times g$ for 20 min, and $3,000 \times g$ for 5 min. The result of centrifugation at $21,880 \times g$ for 20 min was defined as 100%. As shown in **Figure for revise 3**, maleate concentration in 4 different conditions was almost the same. Thus, the separation condition would not affect maleate concentration.

Figure for revise 3 | Relative maleate concentration of culture supernatant of CMtS101 after 72 h in 4 different separation conditions.

Comments 10

Is there a reference for the statement on p 5: “however, microorganisms capable of degrading aromatic compounds do not have the ability to endogenously produce gentisate from sugars or other

renewable carbon sources via the central metabolic microbial pathways such as the glycolysis pathway or pentose phosphate pathway”.

Response

In our knowledge, there are no reports about the production of gentisate using microorganisms capable of degrading aromatic compounds. It may be difficult to show “nothing to be reported” using any references.

References for revise

1. Noda, S., Shirai, T., Oyama, S. & Kondo, A. Metabolic design of a platform *Escherichia coli* strain producing various chorismate derivatives. *Metab. Eng.* **33**, 119–129 (2016). doi: 10.1016/j.ymben.2015.11.007.
2. Weiner, M., Tröndle, J., Albermann, C., Sprenger, G.A. & Weuster-Botz, D. Improvement of constraint-based flux estimation during L-phenylalanine production with *Escherichia coli* using targeted knock-out mutants. *Biotechnol. Bioeng.* **111**, 1406–16 (2014). doi: 10.1002/bit.25195.
3. Song, C.W., Lee, J., Ko, Y.S. & Lee, S.Y. Metabolic engineering of *Escherichia coli* for the production of 3-aminopropionic acid. *Metab. Eng.* **30**, 121–129 (2015). doi: 10.1016/j.ymben.2015.05.005.
4. Song, C.W., Kim, D.I., Choi, S., Jang, J.W. & Lee, S.Y. Metabolic engineering of *Escherichia coli* for the production of fumaric acid. *Biotechnol. Bioeng.* **110**, 2025–2034 (2013). doi: 10.1002/bit.24868.

REVIEWERS' COMMENTS:

Reviewer #1 (Remarks to the Author):

The authors responded well to the points I have raised and included new data.

Reviewer #2 (Remarks to the Author):

The authors addressed all my concerns. An important point is that they can distinguish fumarate from maleate (as shown in Figure for revision 2). The labeling experiments provide conclusive evidence that maleate is produced from glucose. I have no further points. Very nice work.

REVIEWERS' COMMENTS:

Reviewer #1 (Remarks to the Author):

The authors responded well to the points I have raised and included new data.

Response

Thank you for your comment.

Reviewer #2 (Remarks to the Author):

The authors addressed all my concerns. An important point is that they can distinguish fumarate from maleate (as shown in Figure for revision 2). The labeling experiments provide conclusive evidence that maleate is produced from glucose. I have no further points. Very nice work.

Response

Thank you for your comments.